# A new surface meltwater routing model for use on the Greenland Ice Sheet surface

Kang Yang[1,2,3], Laurence C. Smith[4], Leif Karlstrom[5], Matthew G. Cooper[4], Marco Tedesco[6], Dirk van As[7], Xiao Cheng[2,8], Zhuoqi Chen[2,8], Manchun Li[1,3]

[1]School of Geography and Ocean Science, Nanjing University, Nanjing 210023, China
[2]Joint Center for Global Change Studies, Beijing 100875, China
[3]Jiangsu Provincial Key Laboratory of Geographic Information Science and Technology, Nanjing 210023, China
[4]Department of Geography, University of California, Los Angeles, California 90095, USA
[5]Department of Earth Sciences, University of Oregon, Eugene, Oregon 97403, USA
[6]Lamont-Doherty Earth Observatory, Columbia University, Palisades, New York 10964 USA
[7]Geological Survey of Denmark and Greenland, Øster Voldgade 10, 1350 Copenhagen, Denmark
[8]State Key Laboratory of Remote Sensing Science, College of Global Change and Earth System Science, Beijing Normal University, Beijing 100875, China

*Correspondence to*: Kang Yang (kangyang@nju.edu.cn)

**Abstract.** Large volumes of surface meltwater are routed through supraglacial internally drained catchments (IDCs) on the Greenland Ice Sheet surface each summer. Because surface routing impacts the timing and discharge of meltwater entering the ice sheet through moulins, accurately modelling moulin hydrographs are crucial for correctly coupling surface energy and mass balance models with subglacial hydrology and ice dynamics. Yet surface routing of meltwater on ice sheets remains a poorly understood physical process. We use high-resolution (0.5 m) satellite imagery and a derivative high-resolution (3.0 m) digital elevation model to partition the runoff-contributing area of Rio Behar catchment, a moderate-sized (~63 km$^2$) mid-elevation (1,207-1,381 m) IDC on the southwestern Greenland ablation zone, into meltwater open-channels (supraglacial streams and rivers) and interfluves (small upland areas draining to surface channels, also called "hillslopes" in terrestrial geomorphology). A simultaneous in-situ moulin discharge hydrograph was previously acquired for this catchment in July 2015. By combining the in-situ discharge measurements with remote sensing and classic hydrological theory, we determine mean meltwater routing velocities through open-channels and interfluves within the catchment. Two traditional terrestrial hydrology surface routing models, the unit hydrograph and rescaled width function, are applied and also compared with a surface routing and lake filling model. We conclude: 1) Surface meltwater is routed by slow interfluve flow (~$10^{-3}$ – $10^{-4}$ m/s) and fast open-channel flow (~$10^{-1}$ m/s); 2) The slow interfluve velocities are physically consistent with shallow, unsaturated subsurface porous media flow (~$10^{-4}$ – $10^{-5}$ m/s) more than overland sheet flow (~$10^{-2}$ m/s); 3) The open-channel velocities yield mean Manning's roughness coefficient ($n$) values of ~0.03 – 0.05 averaged across the Rio Behar supraglacial stream/river network; 4) Interfluve and open-channel flow travel distances have mean length scales of ~$10^{0}$ – $10^{1}$ m and ~$10^{3}$ m respectively; 5) Seasonal evolution of supraglacial stream/river density will alter these length scales and the proportion of interfluves vs. open-channels, and thus the magnitude and timing of meltwater discharge hydrograph received at the outlet moulin. This phenomenon may explain seasonal subglacial water pressure variations measured in a

borehole ~20 km away.  In general, we conclude that in addition to fast open-channel transport through supraglacial streams and rivers, slow interfluve processes must also be considered in ice sheet surface meltwater routing models.  Interfluves are characterized by slow overland and/or shallow subsurface flow, and it appears that shallow unsaturated porous-media flow occurs even in the bare-ice ablation zone.  Together, both interfluves and open-channels combine to modulate the timing and

discharge of meltwater reaching IDC outlet moulins, prior to further modification by en- and sub-glacial processes.

## 1 Introduction

Supraglacial internally drained catchments (IDCs) are hydrologic units on the Greenland Ice Sheet (GrIS) surface that collect and drain surface meltwater through supraglacial stream/river networks to terminal moulins or lakes (Thomsen et al., 1989;Yang and Smith, 2016). IDC spatial and temporal characteristics and processes constrain the location, discharge, lag

times, and total volume of surface meltwater penetrating into the ice sheet via moulins (Banwell et al., 2013;Yang and Smith, 2016;Smith et al., 2017), which in turn influences the timing of surface mass loss, subglacial hydrologic system evolution, and ice flow dynamics on seasonal and shorter-term timescales (Zwally et al., 2002;Sole et al., 2011;Banwell et al., 2013;Andrews et al., 2014;Arnold et al., 2014;Clason et al., 2015;Smith et al., 2015;Banwell et al., 2016).

Previous studies have shown that planform IDC locations and shapes are largely induced by underlying bedrock

controls on ice surface morphology, which is influenced by variations in bed roughness and slipperiness  and the differing transmission of that variability to the ice surface (Lampkin and VanderBerg, 2011;Karlstrom and Yang, 2016;Crozier et al., 2018;Ignéczi et al., 2018). IDC areas that generally increase with elevation due to lower moulin densities at high elevations (Poinar et al., 2015;Yang and Smith, 2016). As a result, high-elevation IDCs can drain non-trivial volumes of meltwater into the ice sheet even where overall melt rates are low (Yang and Smith, 2016;Smith et al., 2017). Analysis of satellite imagery

suggests that by mid-July, over 95% of IDCs in the southwest GrIS drain meltwater into moulins rather than lakes (Fitzpatrick et al., 2014;Smith et al., 2015). The variable sizes and shapes of IDCs influence the timing and magnitude of peak meltwater injection into the ice sheet through these moulins (Smith et al., 2017).

Few studies have examined meltwater surface routing processes through IDCs on the Greenland Ice Sheet ablation zone surface. Therefore, our ability to simulate IDC moulin hydrographs and, by extension, realistic surface-to-bed connections,

remains limited. Most studies to date have used surface melt rates simulated by Regional Climate Models (RCMs) to calculate runoff (hereafter called "moulin discharge") assumed to flow to the IDC's terminal outlet moulin, but without explicit treatment of surface meltwater routing processes and associated time lags (Smith et al., 2017). This can introduce large uncertainty in the timing and magnitudes of meltwater injection into the ice sheet, impacting the accuracy of subglacial hydrology and dynamical ice flow simulations.

Previous studies have addressed this problem in different ways. Clason et al. (2015) used a single-flow direction algorithm to route surface meltwater across the ice surface and accounted for factoring in runoff delays due to snowpack retention; Arnold et al. (1998) developed a distributed surface routing and lake filling (SRLF) model to simulate moulin

discharge. The SRLF was designed for snow- or bare ice-covered IDCs, and has since been used to simulate the effects of up-glacier snowline retreat on meltwater routing (Willis et al., 2002), drive subglacial hydrologic system evolution (Banwell et al., 2013;Banwell et al., 2016), fill supraglacial lakes (Banwell et al., 2012;Arnold et al., 2014), and illustrate ice flow patterns (de Fleurian et al., 2016). Leeson et al. (2012) developed a similar surface meltwater routing model based on

Manning's equation for open-channel flow and Darcy's law for subsurface flow through a porous medium to simulate lake evolution on the GrIS.

As with terrestrial hydrologic models, supraglacial meltwater routing models are highly sensitive to the choices of surface routing scheme, some poorly quantified parameters (e.g., Manning's roughness coefficient $n$, mean flow velocity, lag time-to-peak, near-surface permeability) and input data (e.g., RCM model output, Automated Weather Stations). Karlstrom

et al. (2014) measured and modeled subsurface porous flow in weathered ice to infer near-surface permeability and found it to be considerably smaller than commonly assumed parameterizations. Gleason et al. (2016) measured hydraulic geometries (width, depth, velocity, slope, Manning's roughness coefficient $n$, etc.) of nine supraglacial meltwater channels on the southwest GrIS and found $n$, in particular, to be variable (0.009 − 0.154). Smith et al. (2017) measured a 72-hour *in situ* terminal outlet moulin hydrograph for Rio Behar catchment, a moderately large instrumented IDC (~63 km$^2$). These data

were used to empirically calibrate a simple surface meltwater routing model, the Snyder Synthetic Unit Hydrograph (SUH) for broader application across 799 surrounding IDCs remotely sensed across the GrIS ablation zone (Yang and Smith, 2016). However, SUH is a highly "lumped" routing model meaning it does not distinguish between different physical flow processes path within a catchment (Singh et al., 2014), and thereby cannot differentiate between different meltwater routing processes. Some other more complex SUH methods have also been proposed for terrestrial hydrology but most of those

methods cannot partition physical flow processes either (Singh et al., 2014). For example, the Geomorphic Instantaneous Unit Hydrograph (GIUH) method only focuses on open-channel flow but ignores hillslope flow (Moussa, 2008), which is not suitable for representing meltwater routing on the ice surface.

This study presents a spatially-lumped, process-partitioned meltwater routing model to investigate surface meltwater routing parameters (meltwater travel distance, velocity, and time) in Rio Behar catchment, using the in situ moulin

hydrograph of Smith et al. (2017) for calibration. The lumped spatial domain is the moderate IDC scale (~60 km$^2$). Two traditional terrestrial hydrograph analysis tools, the Unit Hydrograph (UH) and Rescaled Width Function (RWF), are used to characterize IDC meltwater routing at a high spatial resolution (3 m) afforded using high-resolution remotely sensed digital elevation model (DEM) acquired simultaneously with the in situ measurements by the WorldView-1 satellite. RWF offers particular advantages over SUH because it can partition between two different types of flow path (interfluve vs. open-

channel), and quantify their respective mean (or "bulk") meltwater travel velocities within the catchment. The performance of our RWF model is also compared with SRLF to better estimate the performance of RWF and to assess two different routing approaches. The resultant travel distance, velocity, and time are used to characterize two physically different meltwater routing processes on the ice surface, and the implications of seasonal routing evolutions are discussed. We

concluded that a calibrated, high-resolution RWF surface routing model offers good utility for obtaining important meltwater routing parameters and modeling IDC outlet moulin discharge hydrographs on the GrIS bare ice ablation zone.

## 2. Study area

Our study area is the Rio Behar catchment (Smith et al., 2017), a moderate sized supraglacial IDC on the southwest GrIS surface (Fig. 1). It is located in the upper ablation zone, spanning 1207 – 1381 m a.s.l., near the long-term equilibrium line altitude (~1550 m a.s.l.) of this area (van de Wal et al., 2015). In July 2015, the remotely sensed area of the Rio Behar catchment was 63.1 km$^2$ and the mainstem length of its trunk supraglacial river was 13.8 km (Smith et al., 2017). Visual analysis of multi-temporal high-resolution satellite and UAV (drone) images shows that the supraglacial stream/river channel network was highly developed in Rio Behar catchment by 21-23 July 2015, when 93.5% of the surface was bare ice. A detailed description of the Rio Behar catchment and remotely sensed imagery, DEM, and catchment map is provided in Smith et al. (2017). Notably, the IDC exhibits sub-grid scale with respect to RCMs and Ice Sheet Models (ISMs) and is considered dominating surface meltwater routing process on the southwest GrIS surface (Yang and Smith, 2016).

## 3. Data sources

For 72 continuous hours (11:00AM 21 July to 10:00AM 23 July 2015) measurements of meltwater discharge exiting Rio Behar catchment were collected using an Acoustic Doppler Current Profiler (ADCP) in the mainstem supraglacial river (Smith et al., 2017). Hourly simulations of GrIS meltwater production $M$ over the study period were generated using the MAR (Modèle Atmosphérique Régionale) 3.6 RCM (Fettweis et al., 2013). The 20 km MAR grid cells were reprojected to a common 5 km posting and map projection using nearest neighbour resampling. Catchment-mean hourly melt [mm/h] was obtained by clipping MAR grid cells with the remotely sensed Rio Behar boundary (Smith et al., 2017) and summing their corresponding melt values (Fig. 2) , and multiplied by the measured runoff coefficient for the catchment (0.69) to yield units of effective melt $M$' (Smith et al., 2017).

Two catalogs of stereo WorldView-1 (WV1) panchromatic images (spatial resolution 0.5 m) acquired on 18 July 2015were used for detailed mapping of supraglacial hydrologic features (rivers, lakes, and moulins) (Smith et al., 2017). A fixed-wing UAV (Ryan et al., 2015) acquired aerial camera imagery (RGB bands) over the Rio Behar catchment from 20-22 July 2015was used to validate the accuracy of supraglacial stream/river delineations derived from the WV1 image.

A concurrent high-resolution (3 m) DEM was derived from these WV1 stereo images using the open source Ames Stereo Pipeline (ASP) toolkit methods (Shean et al., 2016). We used the 30 m GIMP (Greenland Ice Mapping Project) DEM v2 (Howat et al., 2014) for comparison with WV1 DEM.

## 4. Methods

### 4.1 Remote sensing of Rio Behar supraglacial river network

A supraglacial stream/river network integrates the hydrologic response of an IDC to surface melt from outside the supraglacial channel system (e.g., the "hillslope" in terrestrial hydrology (D'Odorico and Rigon, 2003)) and open-channel flow (Montgomery and Foufoula-Georgiou, 1993). By late July the ice surface of the GrIS ablation zone becomes heavily dissected with very high drainage density (Smith et al., 2015;Yang et al., 2017) and very short distances between open-channels. The term "interfluve" is borrowed from terrestrial fluvial geomorphology and refers to small areas of dissected terrain that slope toward rills or gullies. Interfluves are commonly referred to using the more general term "hillslope", however in terrestrial geomorphology hillslopes can also refer to much large features whereas interfluves are a narrower term used for small upland areas with short runoff distances, typically found on heavily dissected surfaces. It is thus the more appropriate term for use on the ice sheet owing to the high observed stream density and correspondingly short distances between supraglacial open channels. Moreover, on the ice surface, there is no analog of soil creep, which is an important process in terrestrial hillslope geomorphology (Montgomery and Dietrich, 1989;Montgomery and Dietrich, 1992). For these reasons we recommend use of the narrower term "interfluve" instead of the general term of "hillslope", although the mathematical treatments are similar for both.

We delineated actively flowing supraglacial streams/rivers from the 0.5 m panchromatic WV1 image, following the automatic detection method of Yang et al. (2017) and Smith et al. (2017). The concurrent UAV image was used to validate the ability of this 0.5 m WV1 image to capture small streams. Two detection thresholds, one conservative (higher) and one non-conservative (lower) with values of 40 and 5 (out of 255), respectively, were applied separately to create two meltwater masks following Gabor-filtering and path-opening processing of the WV1 image (Yang et al., 2017). The conservative threshold extracted linear features that are confidently classified as open-flow channels with clear channel banks and with high spectral contrast from the surrounding ice (Smith et al., 2017), while the non-conservative threshold extracted all the channel-like features in the image (Yang et al., 2015a). The two resultant river masks therefore represent upper and lower limits for the true distribution of the open-channel supraglacial stream/river network that was actively flowing on Rio Behar catchment when the in situ hydrograph was collected.

### 4.2 High-resolution DEM processing

Extracting an IDC supraglacial stream/river network from a DEM requires assignment of a prescribed location for the catchment outlet (sink). For this study, the topographic depression containing the known location of the terminal outlet moulin was used as the sink; all other small depressions were filled as per Karlstrom and Yang (2016). This partially filled DEM was then used to calculate flow directions and a downstream flow contributing area raster (Karlstrom and Yang, 2016). Finally, a global meltwater contributing area ($A_c$) threshold was used to simulate ice surface drainage networks. In practice,

if $A_c$ is set too large (small), modeled drainage networks will underestimate (overestimate) real-world channel travel distances, and overestimate (underestimate) actual interfluve travel distances (Montgomery and Foufoula-Georgiou, 1993;Yang and Smith, 2016). Therefore, by deliberately varying this parameter we are able to simulate the seasonal evolution of the supraglacial stream/river network, which tends to have lowest drainage density early and late in the melt season (Yang et al., 2015b;King et al., 2016;Yang et al., 2017). This study used a DEM stream burning technique to force the DEM to produce a reliable actively flowing river network (Lindsay, 2016). To burn the WV DEM, elevations of DEM raster pixels that are spatially coincident with our remotely sensed supraglacial map were lowered ("burned") by 1.0 m, thereby forcing routed flow to pass through these accurately mapped supraglacial stream/river channels.

## 4.3 Unit Hydrograph

The unit hydrograph (UH) is a transfer function used to simulate the observed hydrograph ($Q$) at a catchment outlet for a unit input of water supply (e.g. 1 mm, 1 cm, etc.) applied uniformly across the catchment (Dingman, 2015). Effective $M'$ is the fraction of total MAR melt production $M$ that is transported all the way to the terminal outlet moulin, i.e. the remainder after multiplying by the field measured runoff coefficient (0.69) (Smith et al., 2017). $M'$ is used as input to the transfer function, and the resulting simulation is called the "direct hydrograph" (as distinguished from the observed hydrograph). The UH was developed for terrestrial hydrologic applications where precipitation (rain or snow) are the dominant hydrologic inputs (Singh et al., 2014), but is well suited for adaptation on ice sheets by substituting measured or modeled water equivalent from ice melt (Smith et al., 2017), as ice melt is the dominant hydrological input to IDCs in southwest GrIS. Rainfall occasionally occurs during summer in our study area (Van As et al., 2017) but none occurred during our study period.

This allows MAR effective melt ($M'$) to be used as the input data for simulating (routing) the direct hydrograph at the IDC terminal outlet moulin as $Q = M' * UH$, where $*$ is a convolution operator (Dingman, 2015). Smith et al. (2017) used the Collins' method (Collins, 1939) to create a UH specific to the Rio Behar catchment (here called "MAR UH"), using MAR-produced $M'$ and ADCP-measured $Q$ to calibrate the UH.

In the present study, we also derive other UH transfer functions, which can be built from a satellite image or DEM. To do this, the travel time ($t$) for each pixel within the IDC is required. Travel time represents the time needed for water to flow from each pixel to the catchment outlet, and the hourly-binned histogram of this travel time raster thus corresponds to a one-hour UH (Liu et al., 2003). Travel time $t$ can be estimated as $t = L/v$, where $L$ is meltwater flow distance and $v$ is flow velocity. Flow distance $L$ can be calculated from DEMs by assuming that meltwater flows from one pixel to the adjacent pixel having the steepest slope (Karlstrom and Yang, 2016).

## 4.4 Surface routing and lake filling (SRLF) model

SRLF is a distributed, physically based model proposed by Arnold et al. (1998). Model input requirements include DEM elevations and a time series of meteorological forcing data. The model has been widely used for studies of surface meltwater

routing in the GrIS ablation zone (Willis et al., 2002;Banwell et al., 2012;Banwell et al., 2013;Arnold et al., 2014;Banwell et al., 2016;de Fleurian et al., 2016). The SRLF uses Darcy's law to route meltwater flow through snow, and Manning's equation to route meltwater flow over bare ice surfaces. The present study focuses exclusively on the latter, because satellite and UAV mapping revealed the surface of Rio Behar catchment to be virtually all bare ice during the observational period

(Smith et al., 2017).

SRLF can be used to calculate meltwater travel time and thus to create UH. In accordance with Arnold et al. (1998), we calculated a meltwater flow velocity ($v$) for each pixel in the Rio Behar catchment (see Appendix I). To compute meltwater travel time, a cost surface was created as $1/v$, which was then used as one input raster and flow direction raster was used as another input to determine specific flow paths. The output raster gives travel time for each pixel. The SRLF UH was created

by hourly binning the histogram of this travel time raster.

## 4.5 Rescaled Width Function (RWF)

The rescaled width function (RWF) is a conceptual runoff routing model that represents the total flow distance ($L$) in a catchment as a combination of an interfluve (hillslope in terrestrial settings) flow distance ($L_h$) and a channel flow distance

($L_c$), i.e., $L = L_h + L_c$. The RWF is an improved version of the width function (WF). WF does not represent interfluve transport and therefore cannot be used for meltwater routing where interfluve flow is important (see Appendix II). If constant flow velocities are assumed for interfluve ($v_h$) and channel ($v_c$) zones, the travel time ($t$) for each pixel is the sum of interfluve travel time ($t_h$) and channel travel time ($t_c$) (Di Lazzaro, 2009). Consequently, catchment UH can be derived from the travel time, which is renamed the RWFUH (Singh et al., 2014).

Determination of $v_h$ and $v_c$ is challenging because water flow velocities are difficult to measure on a catchment scale, especially for interfluve zones (Moussa, 2008). Although water flow velocities can be predicted theoretically for simple porous flow (Karlstrom et al., 2014) and our more general derivation in Appendix III, the field-measured IDC moulin hydrograph provides a good opportunity to directly calibrate $v_h$ and $v_c$ for a real-world melting ice sheet surface. To achieve this, different combinations of $v_h$ and $v_c$ were used to create RWFUHs. These RWFUHs were then used to simulate direct

hydrographs at the IDC terminal outlet moulin for comparison with the corresponding in-situ moulin hydrograph. To evaluate performance between the simulated vs. observed moulin hydrograph, we used the Nash-Sutcliffe model efficiency (*NSE*) cost function (Nash and Sutcliffe, 1970). The *NSE* was calculated for each RWFUH and compared to the field-measured moulin hydrograph and the "optimal" $v_h$ and $v_c$ defined as the combination that maximize the *NSE*.

## 4.6 Seasonal evolution of the supraglacial stream/river network

By late July, shortwave radiation and air temperatures decline and meltwater production within Rio Behar catchment also declines (Smith et al., 2017). In this study, we assume that low-order streams (i.e., very small tributaries as per Yang et al. (2016)) stop flowing by late July (Yang et al., 2017), higher-order streams/rivers stop flowing by mid-August, and only very large, high-order rivers flow into late August, an assumption supported by remotely sensed supraglacial stream/river maps in Smith et al. (2015) and Yang and Smith (2016). In late July when the supraglacial drainage density is high, fast open-channel transport contributes heavily to meltwater flow routing exiting the IDC. As the supraglacial river network declines and drainage density decreases, interfluve zones expand and contribute more surface area to overland and/or porous media transport process to IDC flow routing.

To test this idea, we simulated a temporal evolution of the supraglacial stream/river networks within the Rio Behar IDC after late July to characterize the impact of drainage density decline on the RWFUH hydrograph. This test consisted of defining a series of $A_c$ thresholds (i.e., 250, 500, 1000, 2500, and 5000 pixels) that were used to create artificial supraglacial drainage networks from WorldView DEMs. $A_c$ indicates the minimum meltwater contributing area required to form a supraglacial headwater stream. If a DEM grid cell exhibits a contributing area larger than $A_c$, a supraglacial stream will form and thereby the grid cell belongs to the open-channel zone. In contrast, if a DEM grid cell exhibits a contributing area smaller than $A_c$, supraglacial stream will not form and thereby the grid cell belongs to the hillslope zone. Larger $A_c$ values will yield larger hillslope zones, whereas smaller $A_c$ values will yield larger open-channel zones. The minimum $A_c$ (250 pixels) was used to simulate a well-developed supraglacial stream/river network, while the maximum $A_c$ (5000 pixels) was used to simulate a poorly-developed stream/river network. Variable $A_c$ values were used to simulate dynamic supraglacial stream/river networks and each $A_c$ value was assumed to last for one week. This sequence of $A_c$ thresholds reasonably mimics the seasonal contraction of the supraglacial stream/river networks after its maximum development in late July (Smith et al., 2015). The resultant drainage networks were then used to calculate moulin discharge (i.e., the direct hydrograph) based on optimal open-channel and hillslope velocities calibrated from RWFUH, to demonstrate the influence of seasonal drainage network evolution on the shape and timing of the discharge hydrograph at the IDC terminal outlet moulin.

## 5. Results

### 5.1 Supraglacial stream/river mapping

In total, 3,381 km of actively flowing supraglacial stream/river lengths were confidently mapped (conservative threshold) within Rio Behar catchment, yielding a drainage density of 53.6 km/km$^2$ with water bodies covering 8.2% of the catchment surface. Applying the non-conservative detection threshold, 10,829 km of supraglacial stream/river lengths were mapped, yielding a higher drainage density of 164.3 km/km$^2$ with water bodies covering 24.1% of the ice surface. These bounding

estimates suggest that 76 – 92% of the ice surface area consisted of interfluve zones with the remainder as supraglacial ponds, lakes, rivers, and streams.

Four test sites were selected to further illustrate the performance of conservative and non-conservative stream/river channel detections (Fig. 3). Sites 1 through 3 are located near the three main tributaries of the Rio Behar catchment where supraglacial streams/rivers are very well developed. Site 4 is located upstream, within an area where supraglacial streams are relatively sparse. Applying the conservative detection threshold delineated supraglacial streams/rivers clearly identifiable in the WV1 image, but narrower channel-like (dark linear but not well-channelized) features among those streams/rivers were missed. In contrast, the non-conservative threshold captured these small features with the resultant streams/rivers being very well-developed and having higher drainage density. Visual inspection of the 0.3 m UAV images (RGB bands) reveals that supraglacial channels mapped with 0.5 m resolution WV satellite imagery capture nearly all conservative channels that can be discerned in 0.3 m UAV camera imagery (Fig. 3). However, numerous smaller non-conservative supraglacial streams among large supraglacial rivers were not delineated. Therefore, we conclude that automatically mapped streams/rivers accurately estimate the minimum and maximum extents of channel and interfluve zones in the Rio Behar catchment.

## 5.2 Interfluve and open-channel travel distances

Meltwater travel distance rasters were calculated for each data source and processing approach (Table 1). Fig. 4 shows the interfluve and channel travel distances with conservatively mapped rivers and burned WV DEM. The resultant mean channel travel distance is $7.1 \pm 4.0 \times 10^3$ m, while the resultant mean interfluve travel distance is $19.7 \pm 30.9$ m. This signifies that in Rio Behar catchment during our study period, meltwater travel distances through open channels were ~3 orders of magnitude longer than travel distances through interfluves.

This mean interfluve distance we estimate for Rio Behar catchment is larger than the $0.5 - 5$ m values reported for the Juneau Icefield (Karlstrom et al., 2014), and the $9.0 \pm 3.4$ m interfluve distance reported for another low-elevation IDC of southwest GrIS (McGrath et al., 2011). However, their interfluve distances were calculated as the nearest distance from a interfluve point to its adjacent channel rather than following the topographic flow direction calculated from DEMs, as in Karlstrom et al. (2014) and McGrath et al. (2011) (although for small slopes the two approaches should yield similar results). River detection thresholds significantly impact interfluve distances because higher-density distributed meltwater channels lead to smaller interfluve distances. If the non-conservative river detection threshold is used, the mean interfluve distance calculated from our burned WV DEM is $6.7 \pm 15.0$ m, closer to these previously reported estimates.

## 5.3 Interfluve and open-channel travel velocities

The optimal RWF-calibrated mean open-channel velocity $v_c$ is on the order of $10^{-1}$ m/s, while the optimal mean velocity $v_h$ for interfluves is on the order of $10^{-3} - 10^{-4}$ m/s (Table 1 and Fig. 5). Both conservative and non-conservative approaches quantify open-channel velocities as $v_c = 0.3 - 0.5$ m/s, which are similar to previous field-measured values of $0.25 - 0.5$ m/s

in small supraglacial streams (width < 0.5 m) at the Juneau Icefield (Karlstrom et al., 2014), and 0.35 m/s measured for a small supraglacial stream (0.2 m wide) at the southwest GrIS (Gleason et al., 2016). It is slower than faster velocities (0.2 – 9.4 m/s) measured in large (>10 m) supraglacial rivers (Smith et al., 2015;Gleason et al., 2016). The relatively low $v_c$ = 0.3 – 0.5 m/s quantified for an entire catchment suggests that small, relatively slow-flowing supraglacial streams which are vastly more numerous than large mainstem supraglacial rivers dominated the mean RWF open-channel velocity which is a "bulk" velocity averaged over the entire IDC (Appendix IV).

The RWF-calibrated optimal interfluve velocity $v_h$ shows larger variation than $v_c$. The range of $v_h$ is interpreted as $v_h$ = 0.2 – 1.5 × 10$^{-3}$ m/s if $NSE$ = 0.9. If $NSE$ = 0.925 is used as the calibration threshold, this optimal $v_h$ range is 0.3 – 1.2 × 10$^{-3}$ m/s. Under the assumption that conservative (non-conservative) rivers over- (under-) estimate interfluve distance, the non-conservative $v_h$ should be considered as the lower $v_h$ limit, while the conservative $v_h$ as the upper $v_h$ limit (Table 1). Although the specific $v_h$ may vary according to different calibration thresholds, Fig. 5 suggests that $v_h$ is on the order of 10$^{-3}$ – 10$^{-4}$ m/s. This finding confirms that meltwater is routed through the Rio Behar catchment by slow interfluve flow (~10$^{-3}$ – 10$^{-4}$ m/s) followed by fast open-channel flow (~10$^{-1}$ m/s).

### 5.4 Interfluve and open-channel travel time

The total supraglacial travel time (i.e., the combination of interfluve and channel travel time) for our study area and period was found to be ~11 hours. This suggests that, on average, a unit of application meltwater across the catchment takes 11 hours to arrive at the terminal outlet moulin. However, this should be distinguished from the "time to peak" which describes the lag time between peak meltwater production and peak runoff entering the IDC terminal outlet moulin (Smith et al., 2017), using real-world inputs of melt production which follow a strongly diurnal cycle. The 11 hours reported here refer to a "unit" response, i.e. the average length of time for an instantaneous pulse of 1 cm of meltwater to drain from the Rio Behar IDC.

The optimal $v_h$ and $v_c$ combinations (which maximize $NSE$) were used to calculate mean meltwater travel time in interfluve ($t_h$) and open-channel ($t_c$) zones. The resultant mean $t_h$ is ~6 hours, while the mean $t_c$ is ~5 hours. This result differs from the results obtained for smaller bare ice catchments (<2 km$^2$), in which interfluve travel primarily controls or even dominates meltwater routing (Arnold et al., 1998;Karlstrom et al., 2014). For small catchments, meltwater channels are short and therefore fast travel time through open-channels is less important than slow travel time in interfluve zones.

### 5.5 Moulin hydrograph simulations

Unit hydrographs (UHs) were created from the meltwater travel time maps (driven by the optimal $v_h$ and $v_c$ combinations in Table 1), allowing direct hydrographs at the Rio Behar IDC terminal outlet moulin to be simulated. Similarly, the SRLF model was applied to the WV DEM and the GIMP DEM to create UHs as well (Fig. 6), allowing direct comparisons with our RWF-based methods. Two contributing area thresholds ($A_c$ = 450 m$^2$ and 90 m$^2$) were applied to model supraglacial

drainage networks with large (~23 m) and small (~9 m) interfluve distances, which were used for comparisons with the conservative and non-conservative image-mapped river networks, respectively (Table 1).

The UHs simulated by four RWF-based approaches, using two burned WV DEMs, and two $A_c$-based WV DEMs, generally capture the overall shape of the MAR UH (with a duration of 25 hours, see Smith et al. (2017) for more details). All of these four RWF-based UHs smooth the MAR UH, signifying that surface routing of meltwater distributes runoff more uniformly over time (Dingman, 2015). The SRLF-based WV DEM UH also performs reasonably well, although its shape is more irregular, similar to the MAR UH. The UH simulated by the 30 m GIMP DEM is different from all the other UHs in that it distributes all of the input meltwater during the first 13 hours, suggesting that this coarse resolution DEM overly "speeds up" the surface meltwater transport (Fig. 6a).

Both RWF-based and SRLF-based UHs simulate the moulin hydrograph well (Fig. 6b). Except for the GIMP DEM SRLF approach, all the other RWF-based and SRLF-based approaches were able to accurately simulate the peak discharges of the observed moulin hydrographs. These approaches also captured the peak time of the first daily hydrograph, whereas a 2 – 4 hour time lag was obtained for the second daily hydrograph. However, because SRLF lacks slow interfluve flow, it routes surface meltwater faster than RWF and distributes all of the input meltwater during the first 20 hours (Fig. 6a). Ignoring the slow interfluve flow affects the performance of SRLF method negatively, yielding an optimal $NSE = 0.8742$, smaller than for the RWF method (Table 1). Moreover, the SRLF-based GIMP DEM hydrograph is considerably different from the observed moulin hydrograph. This suggests that DEMs with a resolution exceeding ~30 m may fail to accurately capture the velocities and time delays of surface meltwater routing.

## 5.6 Performance of conventional DEM-based simulations

The two conventional DEM-based simulations (Methods section 4.2), assuming a large and small value of the threshold $A_c$, yielded similar results as the burned DEM approaches. The supraglacial drainage network simulated by a relatively large $A_c$ (450 m$^2$, equivalent to 50 WV DEM pixels) is similar to the conservative image-mapped streams/rivers, whereas the drainage network simulated by the small $A_c$ (90 m$^2$, equivalent to 10 WV DEM pixels) is similar to the non-conservative image-mapped streams/rivers (Table 1). Depending on this choice of $A_c$ threshold and also $NSE$ threshold we find optimal $v_h$ values ranging from $0.3 - 1.2 \times 10^{-3}$ m/s, which is consistent with the optimal range obtained by using the burned WV DEM ($v_h = 0.2 - 1.5 \times 10^{-3}$ m/s). However, $v_c$ shows large variations in optimal values, ranging from 0.4 to 2.0 m/s because DEM-modeled supraglacial drainage networks do not match very well with remotely sensed river networks (Yang et al., 2015b), especially for small supraglacial streams (King et al., 2016). Consequently, the lower value of $A_c = 90$ m$^2$ is recommended for use during the peak melting period if a high-resolution remotely sensed supraglacial stream/river map is not available.

## 5.7 Seasonal evolution of moulin discharge hydrographs

Seasonal changes in the relative proportion of open-channel vs. interfluve zones substantially alter the timing and magnitude of moulin discharge hydrographs (Fig. 7). If the supraglacial stream/river network is well developed (i.e., has high drainage density) and the interfluve zone is small, large diurnal variations in moulin discharge are simulated. This finding suggests that under well-developed conditions, open-channel travel is particularly important, similar to results reported for the SRLF model (Banwell et al., 2013).

As the melt season progresses, smaller supraglacial streams dry up and their associated open-channel zone shrinks. Consequently, open-channel travel becomes secondary to interfluve travel. Under these conditions, meltwater delivery to the englacial system is further attenuated, yielding smaller diurnal variations (Fig.7). This suggests that in absence of a well-developed supraglacial stream/river network, slow interfluve meltwater transport has a "smoothing" effect on terminal outlet moulin discharge. Similar behavior has been observed or simulated (Arnold et al., 1998;Karlstrom et al., 2014) and is here explicitly modeled using RWF.

## 6. Discussion

### 6.1 Surface runoff delays on the Greenland Ice Sheet

The meltwater travel times quantified in this study confirm non-trivial-runoff delays are caused by at least two fluvial meltwater transport processes operating within the Rio Behar catchment. Such delays have previously been considered as insignificant in studies of small ice surface catchments (Karlstrom et al., 2014). However, even for the moderate-sized (~63 $km^2$) Rio Behar catchment, supraglacial rivers are long (>10 km long mainstem), meaning meltwater can take several hours to pass through the open-channel network. In much larger (e.g., ~245 $km^2$ reported in Yang and Smith (2016)) IDCs, channel routing delays are even longer. Therefore, the present study reinforces the importance of supraglacial stream/river networks in imparting non-trivial delays on surface meltwater transport as a function of IDC area, shape and stream length (Smith et al., 2017), with a new contribution of considering slow interfluve flow as well as fast open-channel flow.

In contrast to these prior studies, our results suggest that both interfluve and open-channel processes control the timing and magnitude of meltwater transport on the ice sheet. This finding suggests that slow meltwater passage over short distances on interfluves is compensated by fast meltwater transport over long distances through open channels, such that interfluve travel time was roughly equal to channel travel time during the time of the 2015 field experiment. It is possible, therefore, that supraglacial stream/river networks may mimic the classic graded river concept (Kesseli, 1941;Mackin, 1948), with the open-channel flow network developing into a sufficient density to convey available water supply generated on bare ice interfluves.

Left untreated, surface routing delays degrade the utility of using RCM models to estimate inputs of meltwater to the subglacial environment and proglacial zone. Most current RCM models do not provide any surface routing functions to represent transport of runoff over the ice surface to moulins (Van As et al., 2014;Cullather et al., 2016). To the best of our knowledge, MAR is the only RCM model integrating a runoff delay function to distribute runoff over time. This delay function was proposed by Zuo and Oerlemans (1996) and is based on the idea that surface meltwater reaches supraglacial channels sooner where the general ice surface slope is larger. The coefficients in the delay function were calibrated by albedo observations on the ice surface, and Lefebre et al. (2003) updated the coefficients to route meltwater more quickly. Applying the MAR delay function, the resultant runoff delay for the Rio Behar catchment is 8.6 days based on Zuo and Oerlemans (1996) and 7.5 days based on Lefebre et al. (2003). In contrast, the runoff delay obtained using RWF in our study is only ~11 hours, much shorter than these lumped delays.

Van As et al. (2017) built a statistically based supraglacial-to-proglacial delay function to optimally match modeled runoff with observed proglacial river discharge measurements collected in the Watson River, near Kangerlussuaq. Applying this delay function, the runoff delay for the entire glacial and pro-glacial system is 3.9 days for meltwater generated in the Rio Behar catchment. Van As et al. (2017) used a 10-hour smoothing per 100-m elevation bin to represent supra-glacial routing delays, comparable to our 11-hour travel delay calculated for the Rio Behar catchment. Because this purely statistical approach of Van As et al. (2017) is calibrated using time series of in situ proglacial discharge measurements rather than formulas applied to DEMs as per Zuo and Oerlemans (1996) and Lefebre et al. (2003), its close agreement with our RWF routing model lends confidence in its more physically realistic routing scheme. Note that the catchment areas of most IDCs are commonly smaller than one MAR cell (Yang and Smith, 2016;Smith et al., 2017). Therefore, for all but the largest IDCs many of the surface routing delays modeled explicitly here could plausibly be parameterized at the scale of a single large RCM grid cell.

## 6.2 Seasonal evolution of the supraglacial drainage network

Supraglacial stream/river networks undergo a dramatic seasonal evolution from low- to high- to low-drainage density within just 3-4 months (Lampkin and VanderBerg, 2014;Smith et al., 2015), modifying the shape of IDC terminal outlet moulin hydrographs. Fig. 7 suggests that the moulin hydrograph of the Rio Behar catchment will show small diurnal variations at beginning and end of a melt season and large diurnal variations during peak melt season. Note that this differs from the classic signal of alpine glaciers, which tend to display a steadily intensifying diurnal cycle throughout the summer as seasonal snowlines climb to higher elevations (Elliston, 1973). Because this seasonal variation may significantly modulate subglacial water pressure and consequently ice flow velocities, this effect warrants further study for other IDCs and using ice dynamical models.

Very interestingly, moulin discharges simulated by Fig. 7 are qualitatively similar to subglacial water pressure variation measured in a borehole ~20 km away from our catchment (67.201° N, 49.289° W; Fig. 7 in Wright et al. (2016)). Although

these field-measured subglacial water pressures were obtained during 2011, they show similarly large diurnal variations during late July, smaller diurnal variations during early August, and very small diurnal variations around late August. This may indicate a direct control of seasonally varying surface meltwater routing on subglacial water pressure, which in turn impacts subglacial pathway evolution and ice flow dynamics (Banwell et al., 2013;Wright et al., 2016). Meanwhile, the subglacial drainage network is well developed in late summer (August) (Andrews et al., 2014) so this may also contribute to the small diurnal of subglacial water pressure.

The SRLF model is the most commonly used model for simulating meltwater delivery to moulins and our results suggest it performs well for surface hydrologic conditions similar to those in our study IDC (Banwell et al., 2013;Banwell et al., 2016). More recently, the Synthetic Unit Hydrograph (SUH) has been advanced as a simple way to model the magnitude and timing of moulin runoff based on remotely sensed IDC area, shape, and main-stem stream length (Smith et al., 2017). However, neither SRLF nor SUH considers the seasonal evolution of supraglacial stream/river networks. Owing to its flexibility for partitioning interfluve and open-channel zones and their changing ratios over time (Mutzner et al., 2016), the RWF provides a good opportunity for improved representation of both interfluve and open-channel processes, and their evolution over space and time.

The RWF model is calibrated here using a single in situ supraglacial river discharge hydrograph, so the obtained optimal meltwater routing velocities can thus only be confidently attributed to one particular IDC (Rio Behar catchment) during one particular study period (21-23 July 2015). However, other bare-ice IDCs surrounding Rio Behar catchment have similar surface conditions during bare-ice conditions, suggesting some transferability of our results. For example, the calibrated open-channel velocity, i.e., $v_c = 0.3 - 0.5$ m/s, can be used to estimate Manning's $n$ ($n = R^{2/3}S^{1/2}/v_c$); if hydraulic radius $R$ is set to 0.035 m (Arnold et al., 1998) and slope $S$ is set to the mean ice surface slope of the Rio Behar catchment, i.e., 0.024, the resultant $n$ is $0.033 - 0.054$, which matches up well with $n = 0.050$ used in Arnold et al. (1998) and $n = 0.035 \pm 0.027$ estimated by Gleason et al. (2016). As such, we submit that our estimated "bulk" averaged value of $n = 0.03 - 0.05$ may be a reasonable estimate for supraglacial streams/rivers under bare ice conditions for use in surface meltwater routing models.

**6.3 Is interfluve meltwater dominated by overland flow or subsurface flow?**

Our results suggest that during late July 2015, surface meltwater in Rio Behar catchment was routed by a combination of slow interfluve flow ($\sim 10^{-3} - 10^{-4}$ m/s) and fast open-channel flow ($\sim 10^{-1}$ m/s). The latter RWF-inferred open-channel flow velocities correspond closely with field measurements (Karlstrom et al., 2014;Gleason et al., 2016), lending confidence that our bulk catchment-averaged values are grounded in a real-world process.

Less clear, however, is the physical process governing interfluve flow. The rescaled width function (RWF) method we implement assumes that surface meltwater is routed by two distinct flow processes, i.e., open-channel flow and interfluve flow. The inferred mean interfluve velocity is $\sim 10^{-3} - 10^{-4}$ m/s, which is 2–3 orders of magnitude smaller than the open-

channel velocities ($\sim 10^{-1}$ m/s). While RWF partitioning does not prescribe a physical transport process operating within interfluves, two likely candidate mechanisms are surface sheet flow and subsurface porous flow.

Sheet flow is overland (over-ice in our case) flow taking the form of a thin, continuous film over relatively smooth surfaces and not concentrated into rills or channels (Mays, 2010). Manning's kinematic solution is generally used to analyze the sheet flow (Mays, 2010) and the meltwater flow velocity can be calculated as $v_s = f(n, M', S, L_h)$, where $n$ is set to 0.05, $M' = 1.7$ cm is daily effective melt in the Rio Behar catchment, $S = 0.024$ is the average catchment slope, and $L_h$ is set to $1 - 100$ m. The resultant $v_s$ is $\sim 3 - 8 \times 10^{-2}$ m/s, which is similar to the terrestrial interfluve velocities (Moussa, 2008;Di Lazzaro, 2009;Singh et al., 2014;Mutzner et al., 2016) but still $1 - 2$ orders faster than the ice surface $v_h$ quantified in this study.

This discrepancy suggests that interfluve transport is most likely controlled by sub-surface meltwater flow, i.e., porous media flow. A fully saturated Darcy's law has been used in Arnold et al. (1998) and Leeson et al. (2012) (among many others) to describe meltwater routing on firn/snow surfaces. However, to our knowledge, all models of meltwater routing over bare ice assume ice is impermeable and that Darcy's law is therefore not applicable (Arnold et al., 1998;Banwell et al., 2013;de Fleurian et al., 2016). Field studies do reveal that the bare ice surface of ablating glaciers is often characterized by a porous ice layer termed "weathering crust" (Müller and Keeler, 1969;Fountain and Walder, 1998;Irvine-Fynn et al., 2011;Stevens et al., 2018), and low density well-developed weathering crust has been observed in bare ice of the Rio Behar catchment (Cooper et al., 2018). Our results suggest that in contrast to current practice, principles of porous-media flow may be applied even in the bare-ice ablation zone if conditions of weathering crust and porous low density bare ice are found.

The classic treatment for water transport through porous media is Darcy's law. Darcy's velocity ($v_d$) is defined as $v_d = kS/\varphi$, where $k$ is hydraulic conductivity [m/s], $S$ is slope [m/m] and $\varphi$ is weathering crust ice porosity; and hydraulic conductivity $k$ is calculated as $k = K\rho_w g/\mu$, where $K$ is absolute permeability [m$^2$], $\rho_w$ is water density [kg/m$^3$], and $\mu$ is water viscosity [ kg/m $\cdot$ s] (Arnold et al., 1998;Leeson et al., 2012). We followed Arnold et al. (1998) to set $\mu = 1.8 \times 10^{-3}$ kg/m $\cdot$ s and Karlstrom et al. (2014) to set $\varphi = 0.1$ for weathering crust ice. Slope $S$ is set as the mean slope (0.024) of the Rio Behar catchment. Near-surface ice permeability is highly uncertain, but applying the $10^{-10} - 10^{-11}$ m$^2$ range estimated in Karlstrom et al. (2014), we estimate Darcy's velocity $v_d$ as $1.3 \times 10^{-4} - 10^{-5}$ m/s, one order smaller than the interfluve velocity $v_h$ we quantified. This implies that interfluve flow was not fully saturated in our study area, at least during the time when the ADCP supraglacial river discharge measurements were collected.

This is consistent with the idea that subsurface flow through permeable weathering crust ice is only partially saturated, except perhaps in regions near channel heads that exhibit many interconnected small lakes and fully saturated slush. Partially-saturated subsurface flow can be described by the Boussinesq equation (Bear, 1972), obtained by combining Darcy's law for porous flow with continuity of water, forced by meltwater recharge due to melting (Karlstrom et al., 2014). The resultant partially-saturated (unconfined aquifer) velocity will be similar or lower than the fully-saturated velocity $v_d$ – as shown in the Appendix III and Fig. 8, reasonable values result in $v_h \sim 10^{-4} - 10^{-5}$ m/s. Because neither of these simple models for porous flow matches the inferred meltwater velocity $v_h = 10^{-3} - 10^{-4}$ m/s in interfluve zones of the Rio Behar catchment, we suspect that multiple physical processes are involved in $v_h$. For example, the combination of a relatively fast

overland flow ($\sim$10$^{-2}$ m/s) and a slower porous subsurface flow ($<$10$^{-4}$ m/s), such as might occur for ephemeral channels on a variably saturated substrate, could explain the larger velocities. We leave mechanistic study of such issues to future work.

## 6.4 Advantages and limitations of RWF

The Surface Routing and Lake Filling (SRLF) model is the first to attempt routing of surface meltwater downslope (Arnold et al., 1998). More recently, the Snyder Synthetic Unit Hydrograph (SUH) was used to derive moulin hydrographs (Smith et al., 2017). Both methods simulate observed moulin hydrographs reasonably well, but they cannot insightfully reveal the physical process of surface meltwater routing. Recently, permeable weathering crust was found on the Greenland bare-ice surface (Cooper et al., 2018), rather than impermeable bare ice as previously assumed (Arnold et al., 1998). For this reason, it may not be appropriate to apply principles of supraglacial open-channel flow everywhere on the ice surface, i.e. subsurface flows may be more suitable for describing meltwater transport in the interfluve (hillslope) areas of higher-elevation ice separating meltwater channels. This reality calls for an easy-to-use, straightforward method to partition ice surface into channel vs. non-channel (i.e. interfluve) flow with each experiencing different physical flow processes. The Rescaled Width Function (RWF) is our proposed solution for this partitioning.

We selected RWF over other SUH methods for the following reasons: 1) most SUH methods do not include interfluve (hillslope) transport and consider only the open channel network on water routing (Singh et al., 2014), whereas RWF includes both hillslope and open-channel flows; 2) RWF is straightforward to implement and couple with remote sensing, requiring only hillslope and open-channel zones as inputs; 3) although RWF is a spatially-lumped model, it can provide catchment-scale meltwater routing velocities, which are crucial for broad-scale understanding of ice surface hydrology. The derived mean open-channel velocity is comparable to field-measured velocities in small supraglacial streams, and the derived hillslope velocity is comparable to simulations of a partially saturated subsurface hydrological model. Therefore, RWF appears to be a simple and useful tool for modeling meltwater routing across broad-scale areas of melting ice.

The central hypothesis of UH theory is that catchment response to rainfall (here, melt production) is linear, i.e., variations in input rainfall/melt change only the ordinates, not duration, of the direct hydrograph (Dingman, 2015). Meltwater routing is also assumed to be fully determinable by the morphometric characteristics of the catchment (here, IDC drainage networks, shape, area, etc.) (Singh et al., 2014). These assumptions of linearity and fixed basin response simplify routing models but also create some limitations. Particularly, hydraulic geometries of meltwater channels are not independent of IDC characteristics but instead vary nonlinearly with channel discharge (i.e., $Q=wdv$, $w=aQ^b$, $d=cQ^f$, $v=kQ^m$). Gleason et al. (2016) suggest that supraglacial meltwater channels primarily accommodate greater discharges by increasing $v_c$ ($m = 0.63 - 0.95$), which is also supported by Brykała (1999) ($m = 0.49$). Thus, at higher discharges open-channel flow velocities will increase nonlinearly. Moreover, $v_h$ is also affected by surface melt dynamics (Leeson et al., 2012;Karlstrom et al., 2014;Karlstrom and Yang, 2016) and spatio-temporal patterns in surface meltwater production surely influence IDC hydrographic responses, just as spatio-temporal variations in rainfall pattern impact terrestrial hydrographic responses

(Nicótina et al., 2008). Therefore, supraglacial IDCs may not respond to surface melt linearly. Future studies should consider varying $v_h$ and $v_c$ based on different surface melt patterns, which should generate variable RWFUHs during a melt season.

In addition to spatio-temporal variations in meltwater input, $v_h$ and $v_c$ surely vary spatially as well. RWF is a spatially-lumped hydrologic model yielding fixed constants of $v_h$ and $v_c$ averaged across the catchment (Table 1). However, in reality, $v_h$ and $v_c$ vary diversely within a catchment (Maidment et al., 1996;D'Odorico and Rigon, 2003). The SRLF model (Arnold et al., 1998) offers spatially varying surface meltwater routing velocities based on ice surface topography much like distributed terrestrial hydrologic models (Maidment et al., 1996;Liu et al., 2003), thus providing a physical based approach to investigate spatially-varying meltwater routing velocities. Therefore, combining RWF and SRLF would be a promising future direction for producing a physically based, spatially-distributed surface routing model for use on ice sheets. As a starting point, RWF-derived interfluve travel velocities could be included in the SRLF model to parameterize meltwater flows through bare, porous low-density bare ice (Cooper et al., 2018), especially for partially saturated subsurface conditions. We leave spatially-distributed routing models for future studies because of two reasons: first, these models need more data inputs and parameters (which are difficult to estimate) than RWF; second, we need to determine what additional scientific value would be gained from more complex models.

Although the four RWF-based approaches presented here (non-conservative mapped rivers, conservative mapped rivers, high $A_c$ threshold, low $A_c$ threshold, see Fig. 6) all simulate the Rio Behar moulin hydrograph reasonably well, it is important to interpret each method from a physically-based standpoint and "get the right answers for the right reasons" (Kirchner, 2006). The four RWF-based approaches all perform well because we calibrated $v_h$ and $v_c$ from field and remote sensing observations and then used the optimal velocity combination to recreate a measured hydrograph. However, these bulk calibrated $v_h$ and $v_c$ values may or may not be reasonable estimates for channel and interfluve velocities more broadly. They were collected using field and remote sensing observations collected during a narrow window of time, from 21-23 July 2015 when snow was gone and the Rio Behar IDC supraglacial stream/river network was highly developed on fully bare ice. We are encouraged that our derived values agree broadly with other field studies (McGrath et al., 2011;Chandler et al., 2013), but additional field investigations are needed to confirm that the mean values of $v_h$ and $v_c$ derived here may be usefully applied to other bare-ice locations on the ice sheet.

### 6.5 Field site and observation recommendation

Selecting of an IDC for field study is logistically challenging and requires careful planning and design. We selected the Rio Behar catchment by considering surface melt intensity, distance to ice edge, distance to automatic weather stations, catchment size and shape, catchment outlet (moulin) conditions, and safety conditions (Smith et al., 2017). Two types of field measurements will be crucial for better understanding of surface meltwater routing process: supraglacial river discharge and subglacial water pressure. Supraglacial river discharge hydrographs can be used to validate the performance of surface

meltwater routing methods, while subglacial water pressure can be used to estimate the hydrological responses of subglacial environments to different supraglacial meltwater inputs (moulin discharge).

In-situ investigation is also necessary to characterize interfluve conditions. Cooper et al. (2018) analyzed the density and hydrological properties of bare, ablating ice away from open channels, by drilling holes into wet bare ice and measuring the subsurface porosity and water infilling rate, properties that cannot be measured from remote sensing. Satellite images are certainly useful for providing preliminary observations for ice surface conditions. For example, Smith et al. (2017) partitioned bare ice and snowpack zones using high-resolution satellite imagery and Ryan et al. (2018) investigated ice surface albedo, surface impurities, and cryoconite holes using higher-resolution UAV images. That said, we are unaware of any remote sensing solution to confirm presence/absence of saturated subsurface weathering crust and its hydraulic conductivity, so field measurements remain essential at present.

## 6.6 Future research directions

It is crucial to couple surface meltwater routing models with subglacial hydrological models to build a complete understanding of surface-to-bed meltwater connections. One path forward would be to use SUH, SRLF, and/or RWF to calculate moulin hydrographs using DEMs of different sources and spatial resolutions, then coupling this output to the Subglacial Hydrology and Kinetic, Transient Interactions (SHaKTI) subglacial hydrology model (Sommers et al., 2018). Doing so would allow derivation of hourly changes in subglacial water pressure in response to different moulin discharge inputs. A logical next step would be to then analyze the potential impact of these varying subglacial water pressures on subglacial hydrologic system evolution and ice flow dynamics. An ultimate objective should be to model the complete surface-to-bed meltwater transfer process by using RCMs to generate surface melt, surface routing to generate moulin discharge hydrographs, and subglacial models to track basal water pressure, subglacial hydrological system evolution, and ice flow.

Crucial ice surface topographic characteristics, such as slope, flow direction, flow length, drainage area, and drainage networks, are scale-dependent. Zhang and Montgomery (1994) illustrated DEM resolution significantly impacts hydrological responses of terrestrial catchment to rainfall, using 2 m, 4 m, 10 m, 30 m, and 90 m DEMs. We suggest that DEM source and catchment geo-morphometry both affect a DEM's capability for simulating meltwater routing on the ice surface. In general, a 100 m or coarser resolution DEM may yield larger offsets in simulating moulin hydrographs compared to a 30 m resolution DEM but the specific offsets need further estimation. Moreover, high-resolution ArcticDEM (Noh and Howat, 2015, 2017) raises prospects for studying meltwater routing in unprecedented detail and it covers the entire Greenland Ice Sheet at present. The ArcticDEM products are now released at 2 m, 10 m, 32 m, 100 m, 500 m, and 1000 m resolution (Release 7, September 2018). Therefore, we recommend using ArcticDEM products in future meltwater routing studies.

A particularly promising area for future work will be incorporating surface routing delays in studies of proglacial discharge, in order to remove the effects of supraglacial delays before interpreting subglacial delays and/or storages from

proglacial river discharge hydrographs. For example, in southwestern Greenland, numerous supraglacial IDCs form on the ice surface each summer and route meltwater into moulins (Thomsen et al., 1989;Banwell et al., 2012;Banwell et al., 2013;Arnold et al., 2014;Yang and Smith, 2016). By integrating RWF surface routing with the lumped supra-/en-/sub-glacial delays obtained from proglacial river studies (Rennermalm et al., 2013;Van As et al., 2014;Van As et al., 2017), subglacial

runoff delays can be better separated from supraglacial delays. The corrected subglacial delays could then be used to better interpret the coupling of surface melt/runoff with subglacial water pressures (van de Wal et al., 2015;Wright et al., 2016), and used more generally to investigate the evolution of subglacial hydrological system over short time scales (Banwell et al., 2013;Andrews et al., 2014). As a simple example, the supraglacial delay we obtain here for Rio Behar catchment using RWF is ~0.5 days (11 hours), whereas the lumped delay obtained from Van As et al. (2017) is 3.9 days. This suggests subglacial

delays contributed ~90% of the total delay between runoff generation on the ice sheet and its appearance in proglacial river discharge at the ice edge.

## 7. Conclusions

Numerous internally drained catchments (IDCs) are distributed on the southwest Greenland Ice Sheet. These catchments collect and drain meltwater through supraglacial stream/river networks to large, terminal outlet moulins, consequently

constraining the timing and discharge of meltwater flowing over the ice surface to moulins that deliver meltwater to discrete locations on the bed. A growing literature is recognizing the non-trivial influence of supraglacial meltwater transport processes on meltwater received at the bed and proglacial zone. This study has investigated surface meltwater routing processes in Rio Behar catchment, a moderate-sized IDC near Kangerlussuaq, using high-resolution satellite and UAV observations and an in situ field-measured supraglacial river discharge hydrograph collected in late July 2015. Key

meltwater routing parameters were quantified using a Rescaled Width Function (RWF) surface routing model which distinguishes fast meltwater transport through supraglacial stream/river channels from slow transport over interfluves, the latter likely involving partially-saturated, near-surface porous-media flow. Our main contribution is thus to partition interfluve vs. open-channel flow in a surface routing model adapted for use on ice surfaces, using remote sensing and sparse in situ measurements. The model includes two terrestrial hydrologic techniques (unit hydrograph and rescaled width function)

in a simple and flexible approach. Its scientific utility includes quantifying runoff delays on the Greenland ice sheet, improved understanding of open-channel versus interfluve water transport on bare melting ice, improved interpretation of proglacial river discharge hydrographs, and improved coupling of climate/SMB datasets with subglacial borehole studies and models of subglacial hydrology and ice dynamics.

**Acknowledgements**

Kang Yang acknowledges support from the National Natural Science Foundation of China (41501452), the National Key R&D Program (2017YFB0504205), and the Fundamental Research Funds for the Central Universities. Laurence C. Smith and Matthew G. Cooper acknowledge the support of the NASA Cryosphere Program (NNX14AH93G) managed by Dr. Thomas Wagner. Leif Karlstrom acknowledges support from the NASA Rapid Response and Novel Research in Earth Science (NNX16AQ56G). Dirk van As acknowledges the support of the Greenland Analogue Project (GAP) and the Programme for Monitoring of the Greenland Ice Sheet (PROMICE). WorldView imagery and geospatial support for this work were provided by the Polar Geospatial Center, University of Minnesota, under NSF PLR awards 1043681 & 1559691. We thank Michael Willis for providing high-resolution WorldView DEMs.

**Appendix I: Parameter settings in the SRLF model**

In the SRLF model, for each bare ice pixel, meltwater flow velocity ($v$) is calculated using Manning's equation:

$$v = R^{2/3} S^{1/2} / n \qquad (1)$$

where $R$ is the hydraulic radius of the meltwater channel, $S$ is ice surface slope calculated from DEM, and $n$ is the Manning roughness coefficient. We used the mean ice surface slope as an approximate for the channel slope because we do not have any in situ small channel slope measurements.

Most supraglacial streams flow in a series of small and sub-parallel channels and thereby Arnold et al. (1998) used a small, constant value of $R = 0.035$ m for all pixels; in this case, if stream width/depth ratio is set to 5.0 (Karlstrom et al., 2014) (4.5 reported in Yang et al. (2016) and 3.4-12.0 in Knighton (1981)) and a rectangle channel cross section is assumed, stream depth is calculated as ~0.049 m, which is very close to field-measured 0.050 m (Karlstrom et al., 2014) and within the range of 0.030-0.400 m reported in Gleason et al. (2016). Moreover, Arnold et al. (1998) assumed Manning's $n$ as 0.05, which is also supported by field measurements (e.g., $n = 0.035 \pm 0.027$ reported in Gleason et al. (2016), and $0.007 - 0.063$ reported in Brykała (1999)). This value is notably larger than the value used in Leeson et al. (2012), i.e., $n = 0.011$, which was derived experimentally for ice by Lotter (1933). We suggest that field-measured $n$ values of supraglacial channels are more accurate than experimental estimation because some plausible mechanisms (e.g., longitudinal cracks that "intersect channels and persist in the bed" (Gleason et al., 2016), cryoconite pitting, and variable ice bed forms) can increase bed roughness and thereby yield high $n$ values.

**Appendix II: Width function**

Width Function (WF) is a widely used hydrologic modeling method in terrestrial catchment studies and provides an easy-to-use, spatially-lumped approach to quantify the influence of the river network geomorphology on the hydrologic response of a

catchment (D'Odorico and Rigon, 2003). The WF of a catchment is computed as the number of pixels located at a given distance from the outlet following the river network ($L_c$), normalized by the total number of pixels belonging to the river network (Singh et al., 2014). If a constant water flow velocity ($v_c$) is assumed, travel time $t$ can be calculated for all the channel pixels, i.e., $t = L_c/v_c$, and consequently catchment UH can be derived, which was named as WFUH (Singh et al.,

2014). The WFUH performs well for large terrestrial catchments in which the travel time across the interfluve zone is negligible with respect to that in the river network (D'Odorico and Rigon, 2003). However, on the ice surface, interfluve processes are important (or even dominate) (Arnold et al., 1998;Karlstrom et al., 2014) and thereby should not be excluded.

**Appendix III: Boussinesq approximation to porous flow**

Velocity of unchannelized subsurface flow may be estimated through the Boussinesq approximation to porous flow in an

unconfined aquifer (geometry defined in Fig. 8), for which Darcy's Law combined with continuity yields a second order ordinary differential equation (Bear, 1972) that may be solved to find the steady state solution for subsurface water height $h$ as a function of distance $x$ away from a stream:

$$h(x) = \left[ h_0^2 - \frac{x}{L}(h_0^2 - h_L^2) + \frac{M}{k}(L - x)x \right]^{1/2} \tag{2}$$

where $M$ is the rate of melting (m/s) derived from surface energy budget, $h_0$ and $h_L$ are stream depths on either side of the

porous unchannelized zone, and $k = \kappa \rho g/\mu$ is the hydraulic conductivity of the porous ice with $\kappa$ the permeability, $\rho$ ice density, $g$ gravity and $\mu$ the dynamics viscosity of water.

The water divide, where $h(x)$ reaches a maximum, is given by

$$x_w = \frac{L}{2} - \frac{k}{M} \frac{(h_0^2 - h_L^2)}{2L}. \tag{3}$$

For simplicity we will study the case where $h_0 = h_L$ so $x_w = \frac{L}{2}$. The velocity of flow $v_h$ at any point $x$ is given by $v_h =$

$k\,dh/dx$, and the average velocity magnitude on either side of the water divide is

$$v_h = \alpha \left[ \sqrt{1 + \frac{k\,M}{\alpha^2}} - 1 \right] \tag{4}$$

where $\alpha = 2kh_0/L$. If $\frac{k\,M}{\alpha^2} \gg 1$, as we will find in this case, the velocity is well approximated by

$$v_h \sim \sqrt{kM} - \alpha . \tag{5}$$

Although few direct measurements of $\alpha$ exist, we estimate based on field-determined permeabilities of Karlstrom et al.

(2014) that $k \sim 10^{-4} - 10^{-3}$ m/s, $h_0 \sim 0.1 - 1$ m, $L \sim 1 - 100$ m, and $M \sim 10^{-5} - 10^{-4}$ m/s. This range implies $\frac{k\,M}{\alpha^2} \gg 1$, and subsequently $v_h \sim 10^{-5} - 10^{-4}$ m/s.

**Appendix IV: Meltwater channel width distribution of supraglacial stream/river network**

The meltwater channel width was derived using ArcScan tool of the ArcGIS software and width histogram of conservatively mapped supraglacial stream/river networks is shown in Fig. 9. This distribution shows that most supraglacial meltwater channels are narrower than 4 m and the resultant mean channel width is 2.5± 2.0 m, supporting our conclusion that numerous small supraglacial streams control bulk-catchment channel velocity $v_c$. We defined large supraglacial rivers as the features that can be identified by moderate-resolution (10 – 30 m) satellites (e.g., Sentinel-2 and Landsat-8), while small supraglacial streams as the features that can only be identified by high-resolution (0.5 – 2.0 m) satellites (e.g., WorldView-1/2/3/4). It is subjective to determine a threshold width but if required, we recommend 10 m.

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

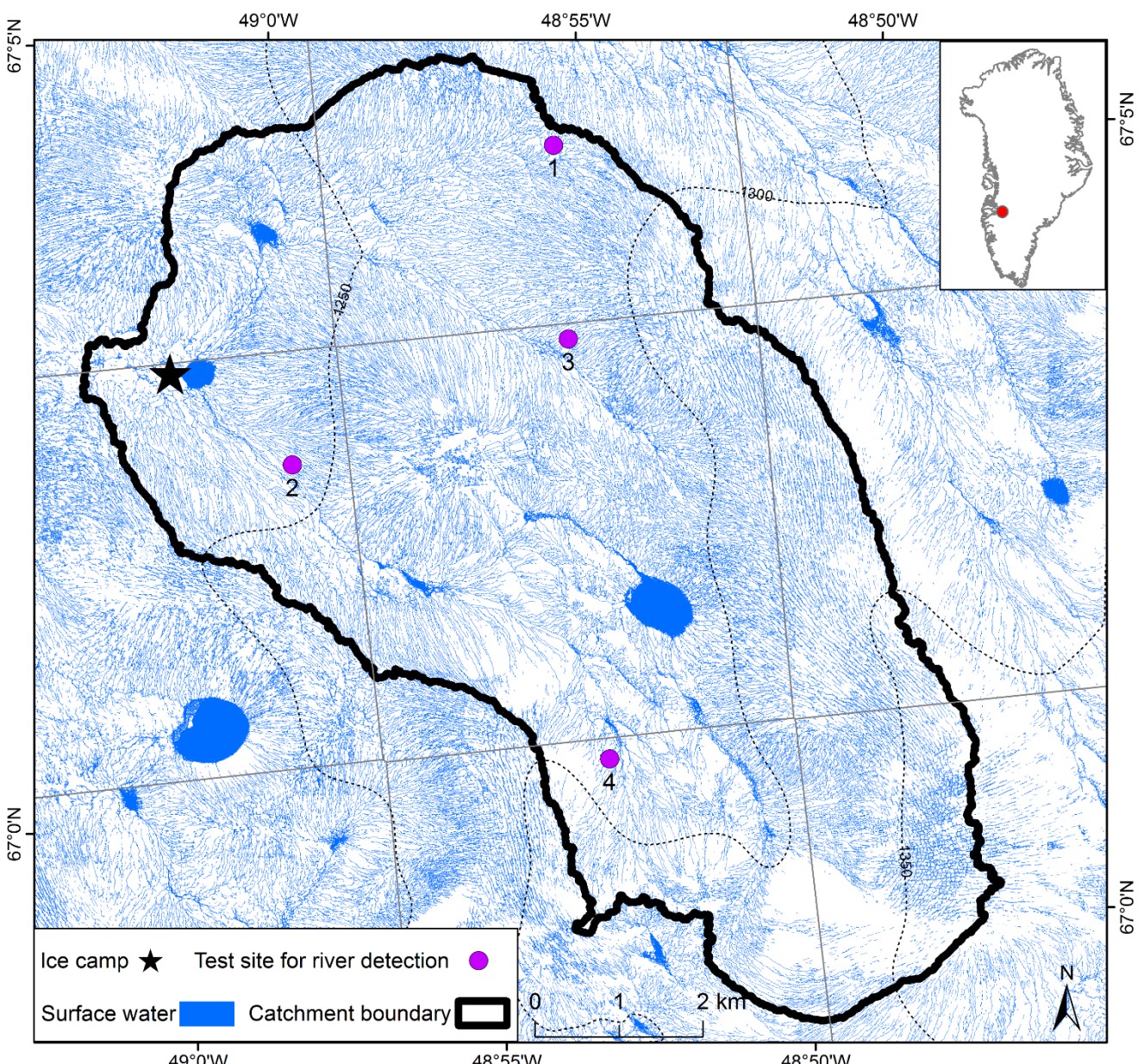

**Figure 1. Rio Behar catchment is a moderate-sized internally drained catchment (IDC) on the southwest Greenland Ice Sheet. A highly developed supraglacial stream/river network was mapped from a high-resolution (0.5 m) panchromatic WorldView-1 (WV1) image acquired on 18 July 2015. The catchment outlet moulin is under the black star and 5 m MAR grid cells are shown in grey**
5  **rectangles. A derivative 3 m resolution DEM is used to delineate the topographic catchment boundary (black line). An in situ hourly hydrograph of supraglacial river discharge collected by Smith et al. (2017) (black star) was used to calibrate a Rio Behar catchment Unit Hydrograph (UH) for the Rescaled Width Function (RWF) and Surface routing and lake filling (SRLF) surface routing models.**

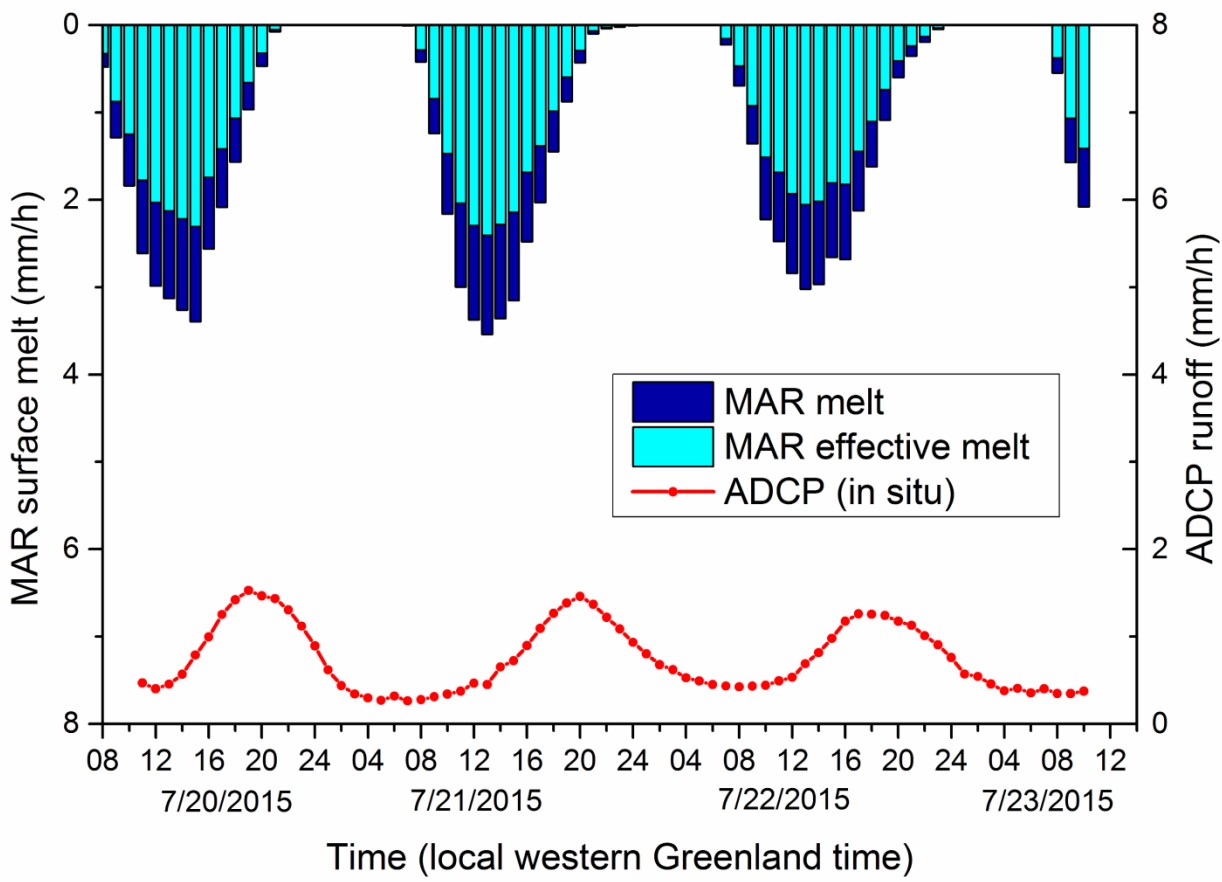

**Figure 2. Within the Rio Behar catchment, MAR regional climate model inputs of melt (at top) are paired with in situ Acoustic Doppler Current Profiler (ADCP) outputs of supraglacial river discharge (at bottom) to calibrate the Unit Hydrograph (UH) surface routing models assessed in this study. "Effective melt" is the fraction of total MAR melt production that is transported all to the Rio Behar IDC terminal outlet moulin (Smith et al., 2017). The UH is calculated using effective melt as inputs and observed runoff as outputs as per (Smith et al., 2017).**

| WorldView-1 image (panchromatic, 0.5 m) | UAV image (RGB, 0.3 m) | Supraglacial river detection conservative | non-conservative |

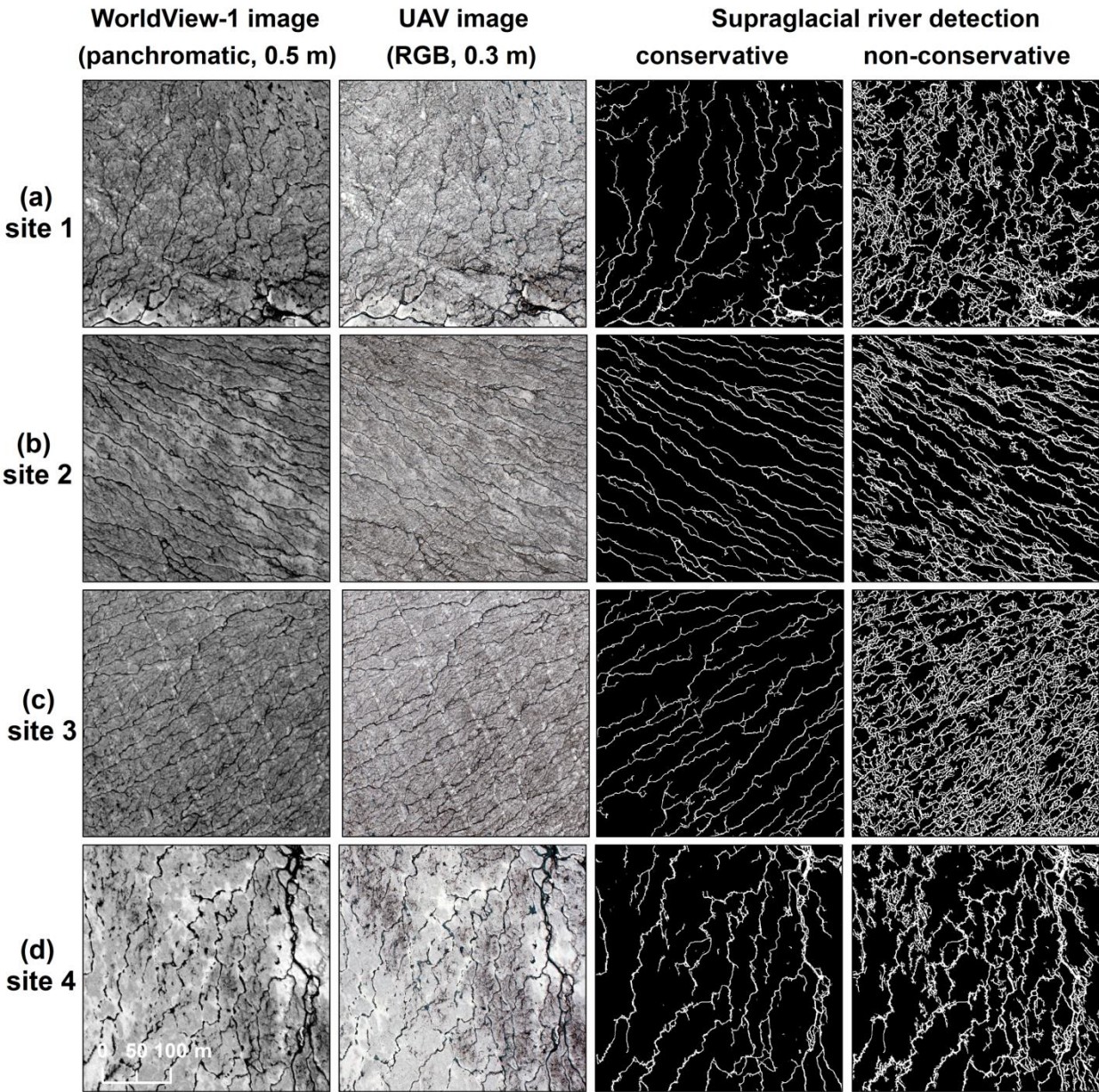

**Figure 3. An 18 July 2015 panchromatic WorldView-1 (WV1) image (spatial resolution 0.5 m, column 1), concurrent UAV imagery (spatial resolution 0.3 m, column 2), and corresponding supraglacial rivers and streams delineated from the WV1 imagery for sites 1-4. Column 3 shows conservative detection of actively flowing channels, while column 4 shows non-conservative detection.**

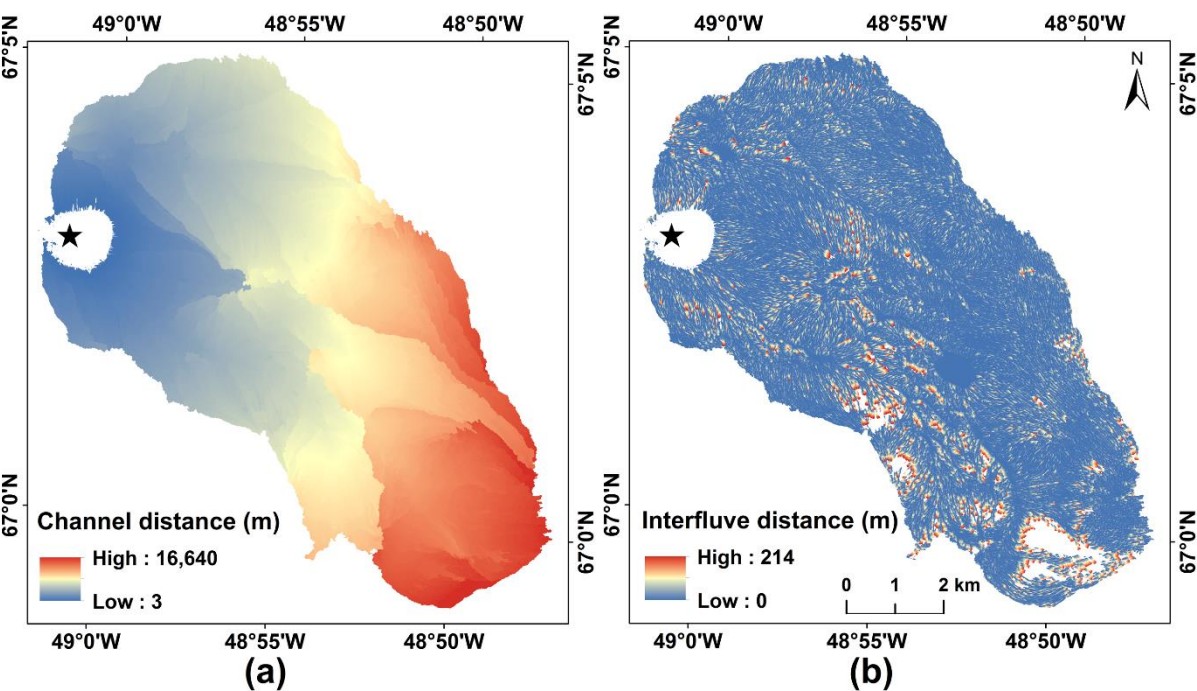

**Figure 4. Meltwater travel distances for (a) open-channels; and (b) interfluves of the Rio Behar internally drained catchment (IDC) as obtained by "burning" the 18 July 2015 WorldView DEM with the conservative remotely sensed active supraglacial stream/river surface drainage network.**

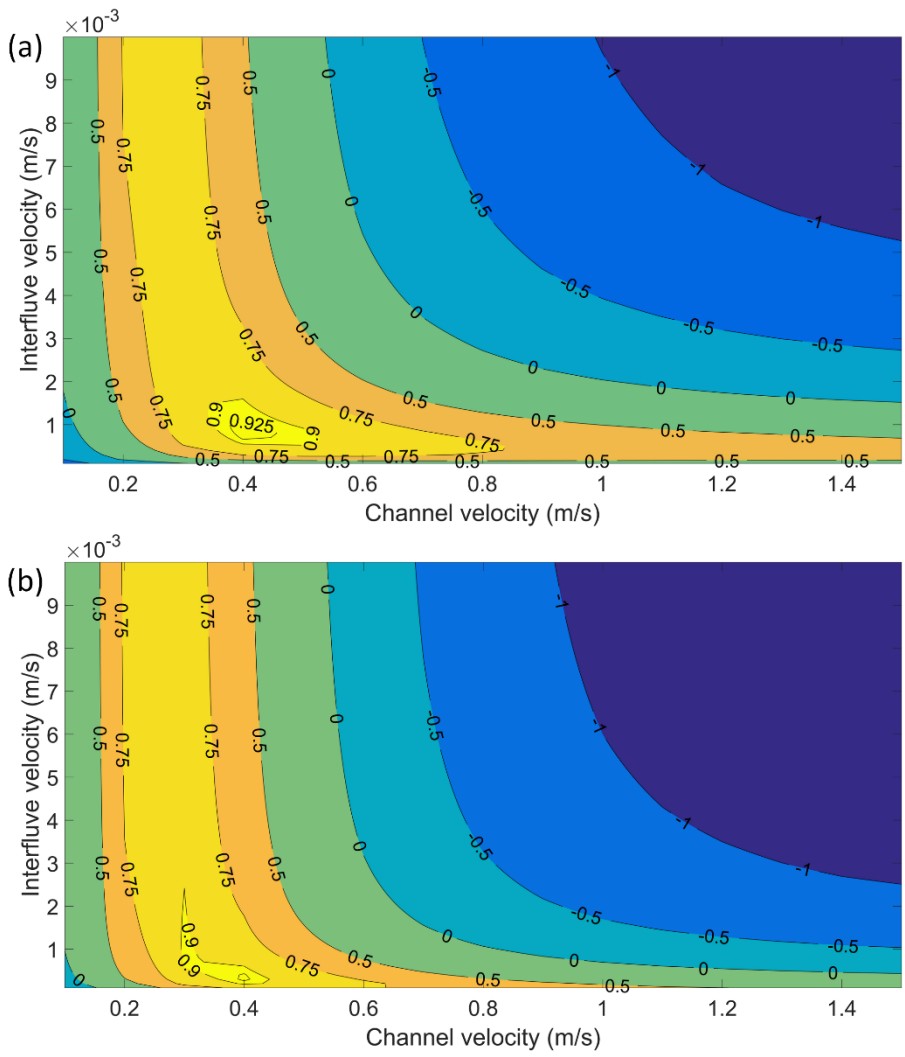

**Figure 5. Calibration of channel and interfluve velocities using (a) conservative and (b) non-conservative supraglacial river delineations. Contour labels show Nash-Sutcliffe model efficiencies (*NSEs*) obtained by applying different combinations of mean interfluve (*v_h*) and open-channel velocities (*v_c*) and comparing with the field-measured hydrograph just upstream of the Rio Behar catchment terminal outlet moulin.**

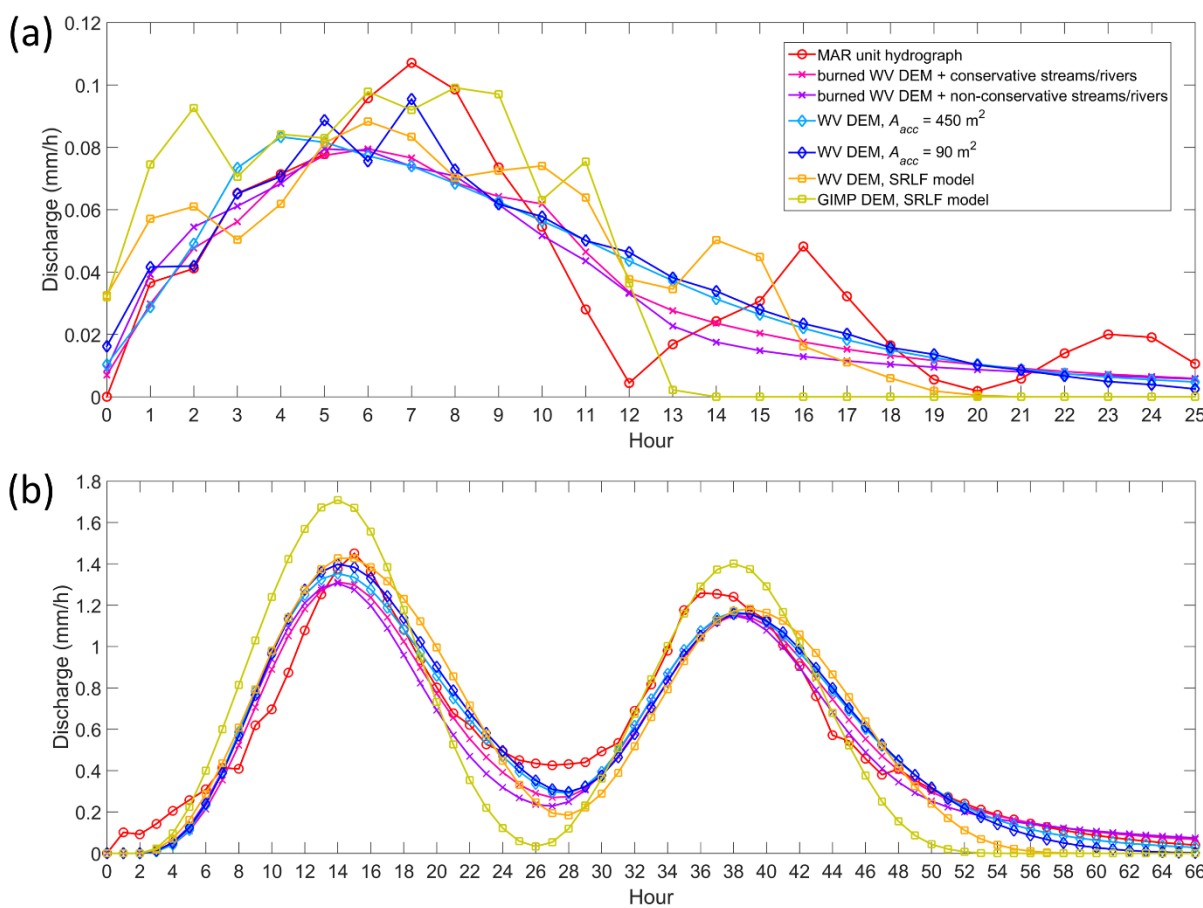

**Figure 6. (a)** Unit hydrographs (UHs) and **(b)** simulated direct hydrographs at the terminal moulin of the Rio Behar catchment, as modeled by different data sources and methods. The MAR UH is calibrated from effective MAR melt and measured supraglacial river discharge (Fig. 2). The Rescaled Width Function (RWF) and surface routing and lake filling (SRLF) UHs are obtained by calibrating open-channel and interfluve velocities to optimally match the MAR UH.

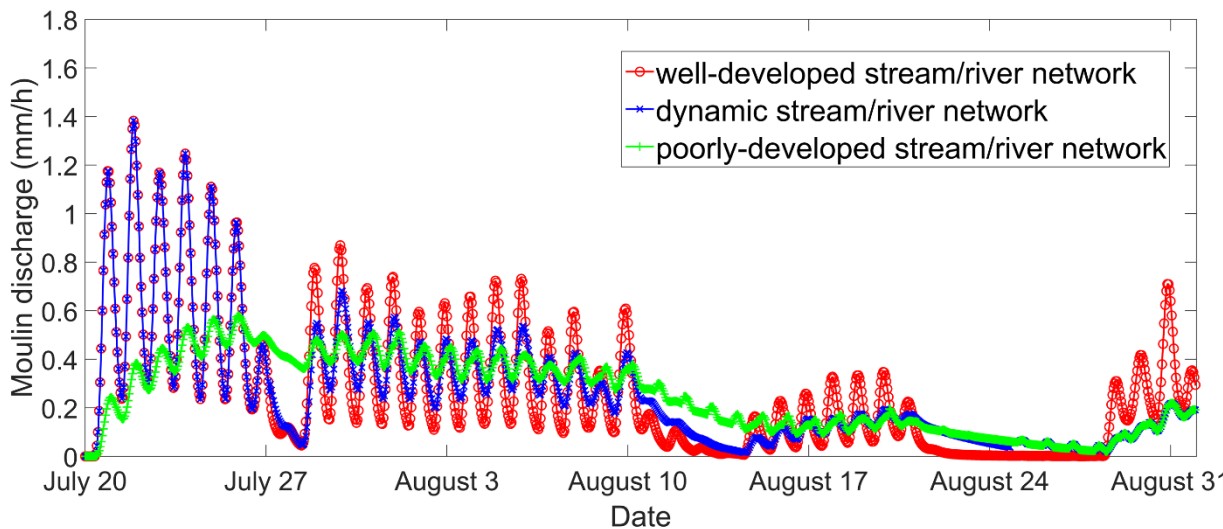

**Figure 7. Simulated Rio Behar catchment discharge at the terminal outlet moulin for 20 July to 31 August 2015 assuming a seasonally evolving supraglacial stream/river network. A series of contributing area thresholds ($A_c$) of 250, 500, 1000, 2500, and 5000 pixels is used to mimic an evolving supraglacial stream/river drainage network from the 18 July 2015 WorldView DEM. The minimum $A_c$ (250 pixels) is used to simulate well-developed river networks, while the maximum $A_c$ (5000 pixels) is used to simulate poorly-developed river networks. Variable $A_c$ values are used to simulate dynamic river networks, which are considered best capturing realistic seasonal evolution. Drainage density is lowest in early and late season, and highest in July. The diurnal variations of moulin discharge are strongly controlled by supraglacial stream/river network patterns.**

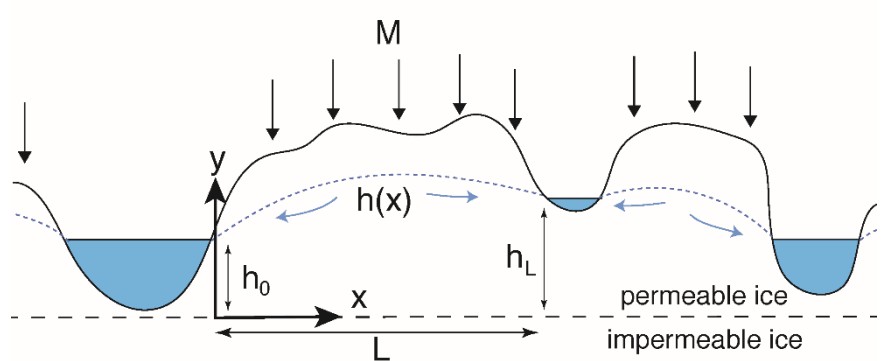

**Figure 8. Schematic diagram of subsurface meltwater porous flow through weathering crust and permeable near-surface, low density ice to estimate interfluve transport speed $v_h$. Melt generated at the near surface at rate M percolates through porous ice, supplying a water table that transports melt downhill towards streams at heights $h_0$ and $h_L$ above a layer of impermeable ice (dashed line).**

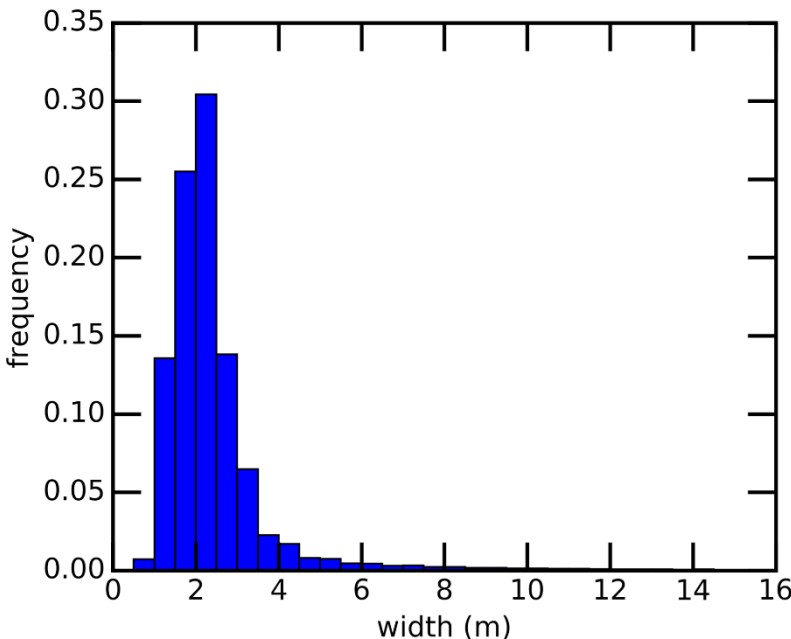

**Figure 9. Channel width histogram of conservatively mapped supraglacial stream/river networks in Rio Behar catchment. Most supraglacial meltwater channels are narrower than 4 m with a mean width of 2.5 ± 2.0 m, confirming that numerous small supraglacial streams dominate the bulk-catchment average channel velocity $v_c$.**

**Table I. Surface meltwater travel distances, velocities, and times in the Rio Behar catchment for July 2015.**

| Input data sources | Mean $L_c$ ($10^3$ m) | Mean $L_h$ (m) | Optimal $v_c$ (m/s) | Optimal $v_h$ ($10^{-3}$ m/s) | Mean $v$ | NSE | Mean $t_c$ (hour) | Mean $t_h$ (hour) |
|---|---|---|---|---|---|---|---|---|
| **Burned WV DEM, conservative threshold** | $7.1 \pm 4.0$ | $19.7 \pm 30.9$ | 0.4 | 0.9 | - | 0.9443 | $4.9 \pm 2.8$ | $6.1 \pm 9.5$ |
| | | | 0.4 | 0.7-1.2 | - | >0.9250 | | |
| | | | 0.3-0.5 | 0.5-1.5 | - | >0.9000 | | |
| **Burned WV DEM, non-conservative threshold** | $7.5 \pm 4.4$ | $6.7 \pm 15.0$ | 0.4 | 0.3 | - | 0.9324 | $5.2 \pm 3.1$ | $6.2 \pm 13.9$ |
| | | | 0.4 | 0.3-0.4 | - | >0.9250 | | |
| | | | 0.3-0.5 | 0.2-2.0 | - | >0.9000 | | |
| **WV DEM, $Ac$ = 50 pixels** | $6.8 \pm 3.8$ | $22.8 \pm 22.2$ | 0.8 | 0.9 | - | 0.9396 | $2.4 \pm 1.3$ | $7.0 \pm 6.9$ |
| | | | 0.6-1.5 | 0.8-1.0 | - | >0.9250 | | |
| | | | 0.5-2.0 | 0.7-1.2 | - | >0.9000 | | |
| **WV DEM, $Ac$ = 10 pixels** | $6.8 \pm 3.8$ | $8.7 \pm 9.0$ | 0.5 | 0.5 | - | 0.9362 | $3.8 \pm 2.1$ | $4.8 \pm 5.0$ |
| | | | 0.5-0.6 | 0.4-0.5 | - | >0.9250 | | |
| | | | 0.4-0.8 | 0.3-0.6 | - | >0.9000 | | |
| **WV DEM, SRLF method** | - | - | - | - | $0.3 \pm 0.1$ | 0.8742 | - | - |
| **30 m GIMP v2, SRLF method** | - | - | - | - | $0.3 \pm 0.1$ | 0.7068 | - | - |