# Peer review of "A new surface meltwater routing model for use on the Greenland Ice Sheet surface"

_The Cryosphere, 2018_

## Referee Comment (RC1) · L. King (Referee) · 1 Oct 2018

Review of: Supraglacial meltwater routing through internally drained catchments on the Greenland Ice Sheet surface, submitted to The Cryosphere Discussions by Yang, K. et al. Reviewer comments.

General comments:

Supraglacial catchment hydrology controls the seasonal and daily inputs of surface meltwater to subglacial catchments, thus modulating subglacial channel evolution and, by extension, ice sheet response to surface melt. However, very little is known about

the routing of water within a supraglacial catchment, and a dearth of empirical data hampers efforts to generalize methods for constructing moulin hydrographs. This paper is a meaningful and logical use of a unique in-situ measured moulin hydrograph, which is combined with traditional hydrological theory to infer the distribution of water routing within different spatial process domains in a supraglacial Internally Drained Catchment (IDC). It is largely a methods paper, but also provides insight into the relative importance and roles of the different hydrological spatial domains in routing flow, as well as how this importance varies seasonally with the evolution of the supraglacial drainage network.

The contributions of this paper to the field are:

1. Unique empirical data on moulin hydrology

2. Methodological advancements on moulin discharge derivations that emphasize the importance of considering the different hydrological processes at work within spatially distinct process domains

3. Insights into the importance of the seasonal evolution of different parts of the supraglacial drainage networks on modifying moulin hydrographs

The contributions of this paper might be enhanced in the following ways:

1. Because this paper is largely a methods paper and relies on empirical data that is available for few supraglacial catchments, it would benefit from a discussion of the choice to use the rescaled width function relative to other methods for deriving synthetic unit hydrographs. What are the practical considerations of this method, and why is it best suited for the supraglacial environment? What other morphometrics might be important in influencing water routing in IDCs, and how might the utility of this method vary spatially? Additionally, given that forward progress in constraining the hydrological processes of IDCs is limited by field data, it would be useful if the discussion section laid out a set of key priority areas for field work that would allow us to generalize this
method beyond the current catchment.

2. This paper is situated in a recent proliferation of studies attempting to generate accurate and generalizable approaches to estimating moulin hydrographs. To further emphasize the contribution of this work beyond its methodological scope, the paper would benefit from further considerations of the implications of this study relative to our understanding of the hydrology of the ice sheet (e.g. under what conditions might we expect the distribution of routing between interfluves and channels to have meaningful impacts on subglacial hydrology, how might this vary in catchments of different sizes, at different elevations, etc..).

RECOMMENDATION: This paper is well written and employs a clear and meticulous methodology with careful consideration of its limitations. I recommend publication of this work with some comments as outlined above and additional minor considerations that I outline below.

Specific comments:

Title: The title is not sufficiently descriptive to distinguish the contribution of this paper from prior contributions in this field. Further, I think that 'internally drained catchments' somewhat misrepresents the work given that the focus of this paper is derivation of the daily moulin hydrograph for a specific IDC. I strongly suggest rewording the title to emphasize that this contribution is at a more spatially, hydrologically, and geomorphologically precise scale than prior work in this area.

Abstract, page 1, line 17: Replace 'it' with specific term – accurately modelling moulin hydrographs?

Introduction

Page 2, line 9: IDCs constrain. . . suggest more specific wording, e.g. : IDC spatial and temporal characteristics and processes constrain. . .

Page 2, line 15: citations for underlying bedrock controls. I suggest citing Lampkin and

Vanderberg (2011) who did earlier work on the topic of bedrock controls on supraglacial hydrological features.

Page 2, Line 29: Clason et al. 2015 did attempt to account for some snowpack retention and runoff delay by factoring in runoff delays due to snowpack retention, although not specifically delays due to routing – would be worth mentioning.

Page 3, Line 18: specify that, in this case, the lumped spatial domain is the IDC scale.

Page 3, Line 21: is the 3 m resolution unprecedently high? ArcticDEM is 2 m resolution and has been used by Karlstrom and Yang (2016) and King et al. (2016) for flow routing in supraglacial environments. Additionally, Rippin, Pomfret and King (2015) used UAV-derived DEMs of 10 cm resolution for derivation of supraglacial channels.

3. Data sources Page 4, lines 29 – 32. For full reproducibility, please include method of degradation and spatial filtering algorithm names (e.g. mean filter, median filter, gaussian filter?).

4.3 Unit Hydrograph Page 6, Line 22: move explanation of M' to line 18, first mention of M'. In this section or in the introduction it would be useful to include a brief discussion of what other SUH methods are available (e.g. Geomorphic Instantaneous Unit Hydrograph) and why they were not employed in this case. Given that the focus of this paper is methodological, it would be useful for other researchers, particularly glaciologists without a familiarity with SUH derivations, to get a broader sense of the range of hydrological approaches that might be used in the context of this work as well as in Smith et al., (2017).

4.5 Rescaled Width Function Page 7, Line 20: can constant flow velocities be assumed for interfluve and channel zones? It seems that flow in channels is highly dependent on location within the network (Gleason et al., 2016). This is acknowledged and addressed in the limitations, but I would be interested in seeing a breakdown of the structure of the channel network in order to get a sense of the scale over which flow velocities vary.

The distribution of total (or mean) channel length by stream order, contributing area, or by channel width would be useful. This could be done as a cumulative distribution, for example, and the effect of the seasonal evolution of the drainage network could be included by showing the drainage network breakdown according to variable Ac values.

5.2 Interfluve and open-channel travel distances Page 9, Line 12: travel distances. I was confused about the travel distance comparison for some time, until it became clear to me that your Lc is in km and your Lh is in m. I suggest for the sake of clarity, put the in-text travel distances in the same units, particularly as Figure 4 is in m for both travel distances. I think it will improve the clarity and readability of this section. Overall for the travel distances section, your findings are that the interfluve travel distances are orders of magnitude shorter than the channel travel distance, and surely this is the same regardless of the channel initiation threshold you use (as per Table 1). Rather than justifying the difference between your findings for the conservative threshold and Karlstrom et al. (2014) and McGrath's et al. (2011)'s findings, I would simply state that although your findings for interfluve travel distances (in particular) vary with initiation area and are closer to prior work at the non-conservative river detection threshold, the orders of magnitude difference between channel and interfluve travel time remains effectively unchanged relative to the difference between the two process domains.

5.3 Interfluve and open-channel travel velocities As per my comment above, I would be interested in seeing some breakdown of the relative dominance of channel of different widths or orders. Assuming you have a mask of channel extents, would it be possible to generate a histogram of river widths in the study area? This would provide some context for comparison between your bulk-catchment vc and prior work.

5.5 Moulin hydrograph simulations Although the SRLF-GIMP hydrograph is different, it does not appear to be 'significantly' so. I wonder what the implications of the observable difference are, and whether these implications are significant at scales that affect subglacial channel evolution. Some discussion of the conditions under which this difference in hydrograph simulations might be accentuated would be useful (e.g. small

vs. large basins, etc. . .). Or, perhaps including the volume of water would be useful for context, rather than just the RWFUH. Amplified by the total volume of water collected in this catchment, how significant does this offset become in a physical sense?

6.1 Surface runoff delays on the Greenland Ice Sheet Page 12, Line 28: is the MAR runoff delay a delay due to runoff routing, or a delay in the production of runoff due to melt storage in the snowpack?

Page 13, Line 6: This is interesting.

6.2 Seasonal evolution of the supraglacial drainage network Page 13, Line 25 – 27: Could these variations in water pressure be due to an evolving sub-glacial network that is better able to transport peak diurnal flow in August? How could we disambiguate these processes?

Page 14 – line 11: Do you not have in-channel measurements of width and depth with which to derive R? R should be in units of meters – in which case your R value seems very low (manning's n is not dimensionless, although it is often represented that way). Also, is the slope value of the catchment surface, or the channel slope? It should be channel slope if you are using the manning's equation for open channel flow.

6.3 Is interfluve meltwater dominated by overland flow or subsurface flow? This is a nice discussion of the mechanisms dominating interfluvial water routing.

6.4 Limitations. This section provides a good overview of the limitations of the RWF method. However, I would also like to see some mention of the morphometrics that are not addressed by the RWF, such as drainage network complexity (e.g. the distribution of streams of different orders), and channel and interfluve slope.

Appendix I: R should have units of m (defined as area [L^2] over perimeter [L]). Again, I am not clear on how a constant R of 0.035 m and thus a depth of 0.05 m is used in this case. Is this meant to be catchment-averaged? For the SRLF, which is distributed, one would expect R and depth to change with every pixel, no? Some clarification of

these assumptions is needed.

Figure 3: according to the text, the WV-1 image was acquired on 18 July, and the UAV image on 20-22 July, therefore the images are not concurrent?

References: Gleason, C. J., Smith, L. C., Chu, V. W., Legleiter, C. J., Pitcher, L. H., Overstreet, B. T., Rennermalm, A. K., Forster, R. R. and Yang, K. (2016) 'Characterizing supraglacial meltwater channel hydraulics on the Greenland Ice Sheet from in situ observations', Earth Surface Processes and Landforms, 41(14), pp. 2111–2122. doi: 10.1002/esp.3977.

Karlstrom, L. and Yang, K. (2016) 'Fluvial supraglacial landscape evolution on the Greenland Ice Sheet', Geophysical Research Letters, 43, pp. 2683–2692.

King, L., Hassan, M. A., Yang, K. and Flowers, G. E. (2016) 'Flow Routing for Delineating Supraglacial Meltwater Channel Networks', Remote Sensing, 8(988). doi: 10.3390/rs8120988.

Lampkin, D. J. and Vanderberg, J. (2011) 'A preliminary investigation of the influence of basal and surface topography on supraglacial lake distribution near Jakobshavn Isbrae, western Greenland', Hydrological Processes, 25(21), pp. 3347–3355. doi: 10.1002/hyp.8170.

Rippin, D. M., Pomfret, A. and King, N. (2015) 'High resolution mapping of supraglacial drainage pathways reveals link between micro-channel drainage density, surface roughness and surface reflectance', Earth Surface Processes and Landforms. doi: 10.1002/esp.3719.

Smith, L. C., Yang, K., Pitcher, L. H., Overstreet, B. T., Chu, V. W., Rennermalm, K., Ryan, J., Cooper, M. G., Gleason, C. J., Tedesco, M., Jeyaratnam, J., van As, D., van den Broeke, M., van de Berg, W. J., Noel, B., Langen, P. L., Cullather, R. I., Zhao, B., Willis, M. J., Hubbard, A., Box, J. E., Jenner, B. A. and Behar, A. E. (2017) 'Direct measurements of meltwater runoff and retention on the Greenland ice sheet

surface', Proceedings of the National Academy of Sciences, 114(50), pp. 1–40. doi: 10.1073/pnas.1707743114.

---

## Referee Comment (RC2) · A.A Leeson (Referee) · 29 Oct 2018

This is a very good paper that makes a solid contribution to our understanding of supraglacial hydrology at the process level. It is well written, provides a nice level of detail and uses a really great dataset. It is a clear example of the scientific advances made possible by the availability of very high resolution satellite data.

I found the discussion of interfluve vs channel flow very interesting and it is this in particular which will be of use to others who are interested in modelling surface hydrology at the broader scale e.g. regional or ice-sheet-wide. The major limitation of this paper in this respect is that the scientific findings are somewhat parochial and it is not clear

at present how far they can be applied beyond the Rio Behar catchment. I would also have liked to have seen more consideration of whether the results scale in the context of modelling supraglacial hydrology on a grid with resolution of the order of 100 m or so. For example a sensitivity analysis with respect to DEM resolution.

I agree with the other reviewer that this paper presents a solid methodological basis for studies in other catchments, and indeed the authors themselves present their study as a starting point for further work at the broader scale. I therefore recommend publication subject to the following, mainly minor, comments being addressed.

Throughout: Please add spaces between references

Page 2, line 11: consider adding 'on seasonal and shorter-term timescales'

Page 2, Line 16: add a sentence about basal-surface transmission being dependant on ice thickness (Lampkin and van der Berg, 2011).

Page 4, section 2: add a sentence acknowledging that this is sub-grid scale with respect to RCMs and ISMs

Page 4, line 15: What are the elevations of the MAR cells used here? How does this compare to the 'real' elevation of the catchment?

Figure 1: Overlay the boundaries of the MAR grid cells used in this study.

Section 3: This section spends too much time repeating Smith et al., 2017. Suggest rolling sections 3 and 4 into one and replacing much of the section 3 text with a table indicating which data comes directly from that paper. This would also help to more clearly outline the novel contribution of this work.

Page 4, line 30: 'point clouds' which 'were'

Page 4, lines 28-30: Why did you need to produce this concurrent DEM?

Page 5, lines 8-9: 'Dissected' is a strange choice of words. I'm not sure I understand

what you mean by it.

Page 5, line 29: a 'DEM'

Page 6, line 1: Does this mean that you do not accumulate water into lakes? Is this justifiable?

Page 6, line 3: I don't understand what Ac is and how it is incorporated into simulations. Could you please explain this better?

Page 6, line 24-26: This should probably go into the list of data taken from Smith et al., 2017

Page 8, line 24-26: Perhaps include a comment on the impact on ice albedo.

Page 9, line 3: How do you define 'channel-like'?

Page 9, lines 6-9: Could you use these data to develop a better channel/non-channel classification? From figure 3 it seems to me that the 'conservative' map agrees better with the UAV image.

Page 9, line12: 'mapped rivers' and burned WV DEM.

Page 9, line 30: How do you define 'large'? A threshold width?

Page 10, line 4 and Table 1: What is 'E'? Please explain.

Page 10, lines 8 and 9: I think this is fairly obvious. Suggest rephrase to 'This finding confirms'

Page 10, section 5.4: What is the 'time to peak' in your catchment? Did you look at this? If not, why not?

Page 11, lines 3 and 4: I think this is significant for broader scale studies where use of a WV DEM is impractical. What about grids of the order of 100 m?

Page 11, line 8: Have you tried modifying your SRLF routine to include interfluve flow?

Page 12, line 1: Also earlier in the melt season I expect, i.e. before your study period starts.

Page 12, line 14: Delete repeated 'IDC'

Page 13, lines 12-14: How? Would you need proglacial discharge measurements for each catchment?

Page 15, line 7: Is it possible to characterise surface conditions using your satellite images or would an in-situ investigation be necessary?

Figure 1: Explicitly say that the moulin is under the black star.

---

## Author Response (AR1)

Dear Editor Michiel van den Broeke,

Thank you for your letter on October 29 inviting revisions to our manuscript, "***Supraglacial meltwater routing through internally drained catchments on the Greenland Ice Sheet surface.***" We are pleased to state that we have complied with all of the requests made by the two reviewers. A stepwise, detailed response to all comments is as follows:

**Reviewer #1**

General comments:

Supraglacial catchment hydrology controls the seasonal and daily inputs of surface meltwater to subglacial catchments, thus modulating subglacial channel evolution and, by extension, ice sheet response to surface melt. However, very little is known about the routing of water within a supraglacial catchment, and a dearth of empirical data hampers efforts to generalize methods for constructing moulin hydrographs. This paper is a meaningful and logical use of a unique in-situ measured moulin hydrograph, which is combined with traditional hydrological theory to infer the distribution of water routing within different spatial process domains in a supraglacial Internally Drained Catchment (IDC). It is largely a methods paper, but also provides insight into the relative importance and roles of the different hydrological spatial domains in routing flow, as well as how this importance varies seasonally with the evolution of the supraglacial drainage network.

The contributions of this paper to the field are:

1. Unique empirical data on moulin hydrology

2. Methodological advancements on moulin discharge derivations that emphasize the importance of considering the different hydrological processes at work within spatially distinct process domains

3. Insights into the importance of the seasonal evolution of different parts of the supraglacial drainage networks on modifying moulin hydrographs

The contributions of this paper might be enhanced in the following ways:

1. ("Because this paper is largely a method paper and relies on empirical data that is available for few supraglacial catchments, it would benefit from a discussion of the choice to use the rescaled width function relative to other methods for deriving synthetic unit hydrographs. What are the practical considerations of this method, and why is it best suited for the supraglacial environment? ")

**Reply:** The Surface Routing and Lake Filling (SRLF) model is the first to attempt routing of surface meltwater downslope (Arnold et al., 1998). More recently, the Snyder Synthetic Unit Hydrograph (SUH) was used to derive moulin hydrographs (Smith et al., 2017). Both methods simulate observed moulin hydrographs reasonably well, but they cannot insightfully reveal the physical process of surface meltwater routing. Recently, permeable weathering crust was found on the Greenland bare-ice surface (Cooper et al., 2018), rather than impermeable bare ice as previously assumed (Arnold et al., 1998). For this reason, it may not be appropriate to

apply principles of supraglacial open-channel flow everywhere on the ice surface, i.e. subsurface flows may be more suitable for describing meltwater transport in the interfluve (hillslope) areas of higher-elevation ice separating meltwater channels. This reality calls for an easy-to-use, straightforward method to partition ice surface into channel vs. non-channel (i.e. interfluve) flow with each experiencing different physical flow processes. The Rescaled Width Function (RWF) is our proposed solution for this partitioning.

We selected RWF over other SUH methods for the following reasons: 1) most SUH methods do not include interfluve (hillslope) transport and consider only the open channel network on water routing (Singh, et al., 2014), whereas RWF includes both hillslope and open-channel flows; 2) RWF is straightforward to implement and couple with remote sensing, requiring only hillslope and open-channel zones as inputs; 3) although RWF is a spatially-lumped model, it can provide catchment-scale meltwater routing velocities, which are crucial for broad-scale understanding of ice surface hydrology. The derived mean open-channel velocity is comparable to field-measured velocities in small supraglacial streams, and the derived hillslope velocity is comparable to simulations of a partially saturated subsurface hydrological model. Therefore, RWF appears to be a simple and useful tool for modeling meltwater routing across broad-scale areas of melting ice.

Additional new text has been added to better discuss the choice to use the RWF relative to other methods, as requested (p. 16, lines 13-31).

2. ("What other morphometrics might be important in influencing water routing in IDCs, and how might the utility of this method vary spatially?")

**Reply:** Thanks for this thoughtful comment. From a hydraulic modeling perspective, supraglacial channel width, depth, and stream order all influence meltwater routing in IDCs. However, these parameters are difficult to estimate at a catchment scale. Moreover, we investigated surface meltwater routing from a hydrological modeling perspective and used RWF to estimate spatially-lumped meltwater routing velocities and transport times. As such, other aspects of meltwater channel morphometrics are not included in this study.

Although RWF is a spatially-lumped model, it has already provided catchment-averaged meltwater routing velocities and reasonable moulin hydrographs. These information may be sufficient to build a surface-to-bed meltwater connections at present since subglacial hydrological models are still at their early development stage and require simple moulin inputs. We leave spatially-distributed routing models for future studies because of two reasons: first, these models need more data inputs and parameters (which are difficult to estimate) than RWF; second, we need to determine what additional scientific value would be gained from more complex models.

Additional new text has been added to better discuss the necessity to develop spatially-distributed models based on RWF in future work (p. 17, lines 23-25).

3. ("Additionally, given that forward progress in constraining the hydrological processes of IDCs is limited by field data, it would be useful if the discussion section laid out a set of key priority areas for field work that would allow us to generalize this method beyond the

current catchment.")

**Reply:** We suggest that the catchment-averaged meltwater routing velocities (hillslope and open-channel velocities) can be applied to other ungauged IDCs. The derived open-channel velocity matches well with field-measured discharge of small supraglacial streams, while the derived hillslope velocity matches with simulations of a partially saturated subsurface hydrological model. Therefore, it is promising to employ RWF to study surface meltwater routing in a broader-scale area.

Selecting of an IDC for field study is logistically challenging and requires careful planning and design. We selected the Rio Behar catchment by considering surface melt intensity, distance to ice edge, distance to automatic weather stations, catchment size and shape, catchment outlet (moulin) conditions, and safety conditions (Smith et al., 2017). Two types of field measurements will be crucial for better understanding of surface meltwater routing process: supraglacial river discharge and subglacial water pressure. Supraglacial river discharge hydrographs can be used to validate the performance of surface meltwater routing methods, while subglacial water pressure can be used to estimate the hydrological responses of subglacial environments to different supraglacial meltwater inputs (moulin discharge).

Additional new text has been added to better discuss generalization of RWF beyond the current catchment, as requested (p. 18, lines 3-10).

4. ("This paper is situated in a recent proliferation of studies attempting to generate accurate and generalizable approaches to estimating moulin hydrographs. To further emphasize the contribution of this work beyond its methodological scope, the paper would benefit from further considerations of the implications of this study relative to our understanding of the hydrology of the ice sheet (e.g. under what conditions might we expect the distribution of routing between interfluves and channels to have meaningful impacts on subglacial hydrology, how might this vary in catchments of different sizes, at different elevations, etc..).")

**Reply:** This is a great suggestion that is unfortunately a research question. Answering it effectively will require coupling a surface meltwater routing model with a subglacial hydrological model, which is beyond the scope of this study. One path forward would be to use SUH, SRLF, and/or RWF to calculate moulin hydrographs using DEMs of different sources and spatial resolutions, then coupling this output to the Subglacial Hydrology and Kinetic, Transient Interactions (SHaKTI) subglacial hydrology model (Sommers et al., 2018). Doing so would allow derivation of hourly changes in subglacial water pressure in response to different moulin discharge inputs. A logical next step would be to then analyze the potential impact of these varying subglacial water pressures on subglacial hydrologic system evolution and ice flow dynamics. An ultimate objective should be to model the complete surface-to-bed meltwater transfer process by using RCMs to generate surface melt, surface routing to generate moulin discharge hydrographs, and subglacial models to track basal water pressure, subglacial hydrological system evolution, and ice flow.

Attempting these steps is beyond the scope of the current paper, but we now outline them as a Section 6.6 "Future research directions" (p. 18, lines 19-29).

Sommers, A., H. Rajaram, and M. Morlighem (2018), SHAKTI: Subglacial Hydrology and Kinetic, Transient Interactions v1.0, Geosci. Model Dev., 11(7): 2955-2974.

RECOMMENDATION: This paper is well written and employs a clear and meticulous methodology with careful consideration of its limitations. I recommend publication of this work with some comments as outlined above and additional minor considerations that I outline below.

Specific comments:

5. ("Title: The title is not sufficiently descriptive to distinguish the contribution of this paper from prior contributions in this field. Further, I think that 'internally drained catchments' somewhat misrepresents the work given that the focus of this paper is derivation of the daily moulin hydrograph for a specific IDC. I strongly suggest rewording the title to emphasize that this contribution is at a more spatially, hydrologically, and geomorphologically precise scale than prior work in this area.")

**Reply:** Thanks for this suggestion. We have changed the title into: "A new surface meltwater routing model for use on the Greenland Ice Sheet surface" (p. 1, lines 1-4).

6. ("Abstract, page 1, line 17: Replace 'it' with specific term – accurately modelling moulin hydrographs?")

**Reply:** "it is" has been replaced with "accurately modelling moulin hydrographs are", as suggested (p. 1, line 19).

7. ("Introduction, Page 2, line 9: IDCs constrain. . . suggest more specific wording, e.g. : IDC spatial and temporal characteristics and processes constrain. . .")

**Reply:** Changed as requested (p. 2, line 12).

8. ("Page 2, line 15: citations for underlying bedrock controls. I suggest citing Lampkin and Vanderberg (2011) who did earlier work on the topic of bedrock controls on supraglacial hydrological features.")

**Reply:** The reference to Lampkin and Vanderberg (2011) has been added, as requested (p. 2, line 19).

Lampkin, D. J., and J. VanderBerg (2011), A preliminary investigation of the influence of basal and surface topography on supraglacial lake distribution near Jakobshavn Isbrae, western Greenland, Hydrolo. Process., 25(21): 3347-3355.

9. ("Page 2, Line 29: Clason et al. 2015 did attempt to account for some snowpack retention and runoff delay by factoring in runoff delays due to snowpack retention, although not

specifically delays due to routing – would be worth mentioning.")

**Reply:** This point has been added, as suggested (p. 3, lines 2-3).

10. ("Page 3, Line 18: specify that, in this case, the lumped spatial domain is the IDC scale.")

**Reply:** "The lumped spatial domain is the moderate IDC scale (~60 km$^2$)" has been added, as suggested (p. 3, line 28).

11. ("Page 3, Line 21: is the 3 m resolution unprecedentedly high? ArcticDEM is 2 m resolution and has been used by Karlstrom and Yang (2016) and King et al. (2016) for flow routing in supraglacial environments. Additionally, Rippin, Pomfret and King (2015) used UAVderived DEMs of 10 cm resolution for derivation of supraglacial channels.")

**Reply:** We meant that a high resolution DEM has never been used to simulate surface meltwater routing so it is unprecedentedly high for this particular application. To avoid misleading readers we have now deleted "unprecedented", as requested (p. 3, line 30).

12. ("3. Data sources Page 4, lines 29 – 32. For full reproducibility, please include method of degradation and spatial filtering algorithm names (e.g. mean filter, median filter, Gaussian filter?).")

**Reply:** We used NASA's Open Source Automated Stereogrammetry Software, ASP (Ames Stereo Pipeline), to create high-resolution DEMs using 0.5 m WorldView (WV) images (Smith et al., 2017). We first used the tool "wv_correct" to correct for subpixel camera alignment artifacts in the full-resolution imagery. The second step is to use the mapproject tool in the ASP to project the left and right images onto a lower resolution DEM. We projected on to the GIMP DEM from OHio State and used bicubic interpolation to downsampled WV images from 50 cm to 1 m. Data source section has been shortened as the Reviewer #2 suggested (p. 5, lines 5-7).

13. ("4.3 Unit Hydrograph Page 6, Line 22: move explanation of M' to line 18, first mention of M'. In this section or in the introduction it would be useful to include a brief discussion of what other SUH methods are available (e.g. Geomorphic Instantaneous Unit Hydrograph) and why they were not employed in this case. Given that the focus of this paper is methodological, it would be useful for other researchers, particularly glaciologists without a familiarity with SUH derivations, to get a broader sense of the range of hydrological approaches that might be used in the context of this work as well as in Smith et al., (2017).")

**Reply:** Some other more complex SUH methods have also been proposed for terrestrial hydrology but most of those methods cannot partition physical flow processes either (Singh et al., 2014). For example, the Geomorphic Instantaneous Unit Hydrograph (GIUH) method includes open-channel flow but ignores hillslope flow (Moussa, 2008), which is not suitable for representing meltwater routing on the ice surface. This point has been added to the Introduction section, as requested (p. 3, lines 22-25). See our reply to your comment #1 for

more detail.

14. ("4.5 Rescaled Width Function Page 7, Line 20: can constant flow velocities be assumed for interfluve and channel zones? It seems that flow in channels is highly dependent on location within the network (Gleason et al., 2016). This is acknowledged and addressed in the limitations, but I would be interested in seeing a breakdown of the structure of the channel network in order to get a sense of the scale over which flow velocities vary. The distribution of total (or mean) channel length by stream order, contributing area, or by channel width would be useful. This could be done as a cumulative distribution, for example, and the effect of the seasonal evolution of the drainage network could be included by showing the drainage network breakdown according to variable Ac values.")

**Reply:** Open-channel and interfluve flow velocities vary spatially in IDCs, as the reviewer pointed out. The constant flow velocities we quantified using Rescaled Width Function (RWF) are "bulk" velocity averaged over the entire IDC. These bulk velocities are useful for characterizing the overall pattern of IDC surface meltwater routing but unfortunately their spatial variations are not derivable using the RWF method. However, Yang et al. (2016) showed supraglacial river width and depth both increase with stream order so we might expect open-channel flow velocities to also increase with stream order based on hydraulic geometric characteristics. Gleason et al. (2016) suggested that supraglacial meltwater channels primarily accommodate greater discharges by increasing velocities. Variable open-channel velocities will lead to different IDC hydrological responses to surface melt and thereby variable UHs will be generated. Several spatially distributed routing methods have been proposed in terrestrial hydrology for this purpose (Melesse and Graham, 2004) and these methods are different from RWF's constant velocity assumption.

To better clarify this difference between RWF and other routing models, additional new text has been added in the Section 6.4 "Advantages and Limitations of RWF" (p. 16, lines 13-31).

Melesse, A. M., and W. D. Graham (2004), Storm runoff prediction based on a spatially distributed travel time method utilizing remote sensing and GIS, J. Am. Water Resour. Assoc., 40(4): 863-879.

15. ("5.2 Interfluve and open-channel travel distances Page 9, Line 12: travel distances. I was confused about the travel distance comparison for some time, until it became clear to me that your $L_c$ is in km and your $L_h$ is in m. I suggest for the sake of clarity, put the in-text travel distances in the same units, particularly as Figure 4 is in m for both travel distances. I think it will improve the clarity and readability of this section. Overall for the travel distances section, your findings are that the interfluve travel distances are orders of magnitude shorter than the channel travel distance, and surely this is the same regardless of the channel initiation threshold you use (as per Table 1). Rather than justifying the difference between your findings for the conservative threshold and Karlstrom et al. (2014) and McGrath's et al. (2011)'s findings, I would simply state that although your findings for interfluve travel distances (in particular) vary with initiation area and are closer to prior work at the nonconservative river detection threshold, the orders of magnitude difference between channel and interfluve travel time remains effectively unchanged relative to the difference between the two process domains.")

**Reply:** We have changed the unit of $L_c$ into m, as requested. In Table I, we used small $A_c$ values on purpose to simulate well-developed supraglacial river networks mapped from satellite imagery but this does not mean that other larger $A_c$ values will yield similar results. For example, if $A_c$ is set to 5000 pixels, supraglacial river network will be poorly developed and the resultant hillslope distance ($L_h$) increases to the order of $10^2$ m. Additional new text has been added to better illustrate this point (p. 8, lines 23-27; p. 9, line 25; p. 37, Table I).

16. ("5.3 Interfluve and open-channel travel velocities. As per my comment above, I would be interested in seeing some breakdown of the relative dominance of channel of different widths or orders. Assuming you have a mask of channel extents, would it be possible to generate a histogram of river widths in the study area? This would provide some context for comparison between your bulk-catchment $v_c$ and prior work. ")

**Reply:** RWF is a spatially-lumped meltwater routing model, meaning that a breakdown of spatially explicit channel widths, stream orders, velocities, etc. is not possible within this particular routing model. However, we generated the meltwater channel width histogram of the conservatively mapped supraglacial stream/river networks in the study area as the reviewer suggested (Figure S1). The result shows that most supraglacial meltwater channels are narrower than 4 m and the resultant mean width is 2.5± 2.0 m, supporting our conclusion that numerous small supraglacial streams control bulk-catchment $v_c$. We have added an Appendix section and new Figure S1 to present this additional work, as requested (p. 22, lines 5-12; p. 36, Figure 9).

[Figure]

**Figure S1.** Channel width histogram of conservatively mapped supraglacial stream/river networks in Rio Behar catchment. Most supraglacial meltwater channels are narrower than 4 m with a mean width of 2.5± 2.0 m, confirming that numerous small supraglacial streams dominate the bulk-catchment average channel velocity $v_c$.

17. ("5.5 Moulin hydrograph simulations. Although the SRLF-GIMP hydrograph is different, it does not appear to be 'significantly' so. I wonder what the implications of the observable difference are, and whether these implications are significant at scales that affect subglacial channel evolution. Some discussion of the conditions under which this difference in hydrograph simulations might be accentuated would be useful (e.g. small vs. large basins, etc. . .). Or, perhaps including the volume of water would be useful for context, rather than just the RWFUH. Amplified by the total volume of water collected in this catchment, how significant does this offset become in a physical sense?")

**Reply:** See our reply to comment 4. This is an outstanding research question for future studies, which could be answered by coupling the RWF routing model with a subglacial hydrological model. Additional new text has been added to Section 6.6 "Future research directions" to explain this (p. 18, lines 19-29).

18. ("6.1 Surface runoff delays on the Greenland Ice Sheet Page 12, Line 28: is the MAR runoff delay a delay due to runoff routing, or a delay in the production of runoff due to melt storage in the snowpack?")

**Reply:** To our understanding, the MAR runoff delay is achieved by three empirically calibrated coefficients, without any physical meanings (e.g., runoff routing or snowpack storage). The underlying idea is that meltwater reaches the drainage system quicker when the general surface slope is larger. Specifically, the delay function is:

$$\frac{dW}{dt} = P_w - \frac{W}{t^*}$$

where $t^* = c_1 + c_2 \exp(-c_3 S)$. $P_w$ is "the production of meltwater that does not refreeze, $t^*$ is "the characteristic time-scale for meltwater runoff", and S is surface slope. Zuo and Oerlemans (1996) determined optimal values for $c_1$, $c_2$, and $c_3$ by "optimizing the simulated albedo against the observations" and the resultant values are 1.5, 25, and 140. Lefebre et al. (2003) updated the coefficients to 0.33, 25, and 140 in order to route meltwater more rapidly. In this study, we show that surface meltwater routing is a crucial physical process but ignored by the RCM models. MAR attempts to impose a delay, but the delay function is purely empirical, with no physical basis, unlike the routing models examined in this study.

19. ("Page 13, Line 6: This is interesting.")

**Reply:** Yes, Van As et al. (2017) presented a very interesting way to quantify surface meltwater routing delay at a broad scale, which directly inspired this study.

20. ("6.2 Seasonal evolution of the supraglacial drainage network Page 13, Line 25 – 27: Could these variations in water pressure be due to an evolving sub-glacial network that is better able to transport peak diurnal flow in August? How could we disambiguate these processes?")

**Reply:** Subglacial drainage network is best-developed in August so it may indeed also contribute to the observed diurnal variations in subglacial water pressure. Separating this effect from supraglacial delays will require coupling RWF with a subglacial hydrologic model, allowing different supraglacial meltwater inputs to disambiguate the contributions of supraglacial and subglacial processes. Additional new text has been added to explain this (p. 14, lines 11-13).

21. ("Page 14 – line 11: Do you not have in-channel measurements of width and depth with which to derive R? R should be in units of meters – in which case your R value seems very low (manning's n is not dimensionless, although it is often represented that way). Also, is the slope value of the catchment surface, or the channel slope? It should be channel slope if you are using the manning's equation for open channel flow.")

**Reply:** We only have in-channel measurements of width and depth at one cross-section at the very end of the main-stem supraglacial river.  On the IDC scale, the mean hydraulic radius ($R$) is dominated by small supraglacial streams due to their numerous number. Arnold et al. (1998) used 0.035 m and we think this is a reasonable assumption. For a small supraglacial meltwater channel with width = 0.25 m, depth = 0.05 m, and rectangle shape, the resultant $R$ is 0.035 m. Because we derived slopes from the DEM, they represent ice surface slope rather than channel slope. We used the mean ice surface slope as an approximate for the channel slope because we do not have any *in situ* small channel slope measurements. Additional new text has been added to better explain this (p. 14, line 27; p. 20, lines 18-20).

22. ("6.3 Is interfluve meltwater dominated by overland flow or subsurface flow? This is a nice discussion of the mechanisms dominating interfluvial water routing.")

**Reply:** Thanks. Determination of interfluve meltwater flow types is crucial for understanding surface meltwater routing. A mechanistic study should be conducted in future to further illustrate the related processes (overland flow, fully saturated subsurface flow, and partially subsurface flow).

23. ("6.4 Limitations. This section provides a good overview of the limitations of the RWF method. However, I would also like to see some mention of the morphometrics that are not addressed by the RWF, such as drainage network complexity (e.g. the distribution of streams of different orders), and channel and interfluve slope.")

**Reply:** Because RWF is a spatially-lumped, process-partitioned meltwater routing model it cannot handle spatially-distributed IDC morphometrics and requires assumption of constant velocities for interfluve and open-channel flow. This is actually an advantage of RWF because it allows RWF to partition mean catchment-scale interfluve and open-channel flow velocities and consequently the overall hydrological response of the IDC to surface melt. We agree with the reviewer that the IDC morphometrics (e.g., drainage network complexity and channel and interfluve slope) should be investigated for building a spatially-distributed

meltwater routing model but due to the "bulk" nature of the RWF model it is beyond the scope of this study. A new section explaining this has been added to the paper (p. 16, lines 13-31).

24. ("Appendix I: *R* should have units of m (defined as area [L^2] over perimeter [L]). Again, I am not clear on how a constant *R* of 0.035 m and thus a depth of 0.05 m is used in this case. Is this meant to be catchment-averaged? For the SRLF, which is distributed, one would expect R and depth to change with every pixel, no? Some clarification of these assumptions is needed.")

**Reply:** See our reply to your comment #21. Arnold et al. (1998) used constant *R* and spatially varied slope to obtain spatially varied velocities for IDC pixels. Thereby, SRLF assumes small supraglacial streams (*R* = 0.035 m) develop everywhere on the ice surface and does not consider variations of supraglacial stream/river network morphometrics (e.g., width, depth, and Stream order) so it is meant to be catchment-averaged. Additional new text has been added to better explain this (p. 14, line 27; p. 20, lines 18-20).

25. ("Figure 3: according to the text, the WV-1 image was acquired on 18 July, and the UAV image on 20-22 July, therefore the images are not concurrent?")

**Reply:** Yes, the WV-1 image and the UAV image are not concurrent. Supraglacial stream/river networks are assumed to have been more or less stable during 18-22 July 2015.

**Reply:** We respectfully disagree that the study is parochial. Rescaled Width function (RWF) is a flexible, simple-to-use spatially-lumped meltwater routing model that takes an important step forward by distinguishing between flow characteristics in open-channels versus interfluves. Because open channels and interfluves are ubiquitous in the bare-ice ablation zone of the western Greenland this conceptual advance should be broadly applicable beyond the Rio Behar catchment on similar bare-ice surfaces of western Greenland. We have compared RWF with other SUH methods, demonstrated the impacts of different moulin inputs on subglacial water pressure, and discussed the necessity to develop spatially-distributed meltwater routing models. See our reply to comment 1-3 of Reviewer #1 for more detail.

It is nontrivial to analyze the impact (sensitivity) of DEM spatial resolution on surface meltwater routing. Crucial ice surface topographic characteristics, such as slope, flow direction, flow length, drainage area, and drainage networks, are scale-dependent. Zhang and Montgomery (1994) illustrated DEM resolution significantly impacts hydrological responses of terrestrial catchment to rainfall, using 2 m, 4 m, 10 m, 30 m, and 90 m DEMs. We suggest that DEM source and catchment geo-morphometry both affect a DEM's capability for simulating meltwater routing on the ice surface. In general, a 100 m or coarser resolution DEM may yield larger offsets in simulating moulin hydrographs compared to a 30 m resolution DEM but the specific offsets need further estimation.

Moreover, high-resolution ArcticDEM (Noh and Howat, 2015, 2017) raises prospects for studying meltwater routing in unprecedented detail and it covers the entire Greenland Ice Sheet at present. The ArcticDEM products are now released at 2 m, 10 m, 32 m, 100 m, 500 m, and 1000 m resolution (Release 7, September 2018). Therefore, we recommend using ArcticDEM products in future meltwater routing studies.

We leave DEM resolution sensitivity for future studies because we focus on RWF in this study. RWF can only be conducted on high-resolution (< 10 m) DEMs because DEM spatial

resolution should not exceed hillslope transport distance; otherwise, hillslope transport distance would be significantly overestimated and the resultant hydrograph would be inappropriate (Hancock et al., 2006). Put another way, coarse-resolution DEMs are unable to differentiate between small channels and interfluves, which are exactly the two surface-types that RWF partitions. In the future, we plan to use the newly released ArcticDEM products to better illustrate the impact (sensitivity) of DEM spatial resolution on surface meltwater routing and consequently investigate surface meltwater routing in a broad-scale area.

Additional new text has been added to better highlight the advantage of RWF model and DEM resolution's impact on surface meltwater routing (Section 6.6 "Future research directions", p. 18, lines 19-31, p. 19, lines 1-7).

Hancock, G. R., C. Martinez, K. G. Evans, et al. (2006), A comparison of SRTM and high-resolution digital elevation models and their use in catchment geomorphology and hydrology: Australian examples, Earth Surface Processes and Landforms, 31(11): 1394-1412.

Zhang, W., and D. R. Montgomery (1994), Digital elevation model grid size, landscape representation, and hydrologic simulations, Water Resour. Res., 30(4): 1019-1028.

**2. ("Throughout: Please add spaces between references")**

**Reply:** Spaces have been added between references, as requested (p. 23-27).

**3. ("Page 2, line 11: consider adding 'on seasonal and shorter-term timescales'")**

**Reply:** Added as suggested (p. 2, line 15).

**4. ("Page 2, Line 16: add a sentence about basal-surface transmission being dependent on ice thickness (Lampkin and van der Berg, 2011).")**

**Reply:** Supraglacial drainage patterns are primarily determined by ice surface topography, which is influenced by variations in bed roughness and slipperiness and the differing transmission of that variability to the ice surface. Additional new text has been added to better explain this, as requested (p. 2, lines 18-20).

**5. ("Page 4, section 2: add a sentence acknowledging that this is sub-grid scale with respect to RCMs and ISMs")**

**Reply:** This sentence has been added to Section 2, as requested (p. 4, lines 13-14).

**6. ("Page 4, line 15: What are the elevations of the MAR cells used here? How does this compare to the 'real' elevation of the catchment?")**

**Reply:** We clipped MAR grid cells with the remotely sensed catchment boundary, so the MAR

cell area used to calculate runoff is equal to the true spatial extent and elevation of the catchment (p. 4, lines 20-21).

7. ("Figure 1: Overlay the boundaries of the MAR grid cells used in this study.")

**Reply:** Changed as requested (p. 28, Figure 1).

8. ("Section 3: This section spends too much time repeating Smith et al., 2017. Suggest rolling sections 3 and 4 into one and replacing much of the section 3 text with a table indicating which data comes directly from that paper. This would also help to more clearly outline the novel contribution of this work.")

**Reply:** We suggest that it may be better to make this paper self-contained (i.e., independent from Smith et al. (2017)). If we only show a data table without further illustrations, it will be difficult for readers to understand Figure 1 and 2. Subsequently, readers may be confused by the descriptions of Unit Hydrograph and Rescaled Width Function, which are directly related to the Data section. However, we agree with the reviewer that Data section repeated Smith et al. (2017) too much so we have shortened this paragraph to more clearly outline the contribution of this work, as suggested (p. 4, lines 15-30; p. 5, lines 1-8).

9. ("Page 4, line 30: 'point clouds' which 'were'")

**Reply:** Changed as requested (p. 5, line 6).

10. ("Page 4, lines 28-30: Why did you need to produce this concurrent DEM?")

**Reply:** This concurrent DEM is used to run both RWF and SRLF models.

11. ("Page 5, lines 8-9: 'Dissected' is a strange choice of words. I'm not sure I understand what you mean by it.")

**Reply:** By "Dissected", we mean supraglacial stream/river networks heavily incised via thermal erosion of underlying ice. We borrow the term from terrestrial geomorphology, which often uses the word "dissected" to describe badlands and other landscapes that are rapidly eroding due to fluvial activity.

12. ("Page 5, line 29: a 'DEM'")

**Reply:** Changed as requested (p. 6, line 4).

13. ("Page 6, line 1: Does this mean that you do not accumulate water into lakes? Is this justifiable?")

**Reply:** These small depressions are considered to be either pathway lakes or DEM noise. This

is justifiable, as we can see from the WorldView satellite imagery and the image-mapped supraglacial stream/river network (Figure 1) that surface meltwater produced in the catchment is routed downslope to the catchment outlet (moulin), without accumulating in lakes.

14. ("Page 6, line 3: I don't understand what $A_c$ is and how it is incorporated into simulations. Could you please explain this better?")

**Reply:** $A_c$ indicates the minimum meltwater contributing area required to form a supraglacial headwater stream. If a DEM grid cell exhibits a contributing area larger than $A_c$, a supraglacial stream will form and thereby the grid cell belongs to the open-channel zone. In contrast, if a DEM grid cell exhibits a contributing area smaller than $A_c$, supraglacial stream will not form and thereby the grid cell belongs to the hillslope zone. Larger $A_c$ values will yield larger hillslope zones, whereas smaller $A_c$ values will yield larger open-channel zones. Therefore, a series of increasing $A_c$ values can be used to simulate temporal declining of supraglacial stream/river networks and to create different IDC hydrological responses to declining surface melt inputs. Additional new text has been added to better explain this, as requested (p. 8, lines 23-27).

15. ("Page 6, line 24-26: This should probably go into the list of data taken from Smith et al., 2017")

**Reply:** This sentence explains the approach to create UH so we suggest that it belongs to the method section. We moved it to the previous paragraph to make the logical flow smoother (p. 6, lines 19-21).

16. ("Page 8, line 24-26: Perhaps include a comment on the impact on ice albedo.")

**Reply:** Conservative and non-conservative thresholding are two ways to delineate supraglacial streams/rivers with different confidences. We suggest that ice albedo does not interact with these two thresholds.

17. ("Page 9, line 3: How do you define 'channel-like'?")

**Reply:** By "channel-like", we mean narrow, dark linear but not well-channelized feature in the ice surface image. This point has now been better explained (p. 9, line 15).

18. ("Page 9, lines 6-9: Could you use these data to develop a better channel/non-channel classification? From figure 3 it seems to me that the 'conservative' map agrees better with the UAV image.")

**Reply:** Yes, 0.3 m UAV images can be used to create a higher resolution supraglacial stream/river map. However, the spatial coverage of our UAV images is smaller than that of WorldView satellite images and three UAV image strips obtained from three days would be

needed to cover the Rio Behar Catchment. For consistency, we therefore used the WorldView image to map supraglacial streams/rivers at 0.5 m spatial resolution. The conservative map represents the relatively large supraglacial rivers well so it may visually appear to agree better with the UAV image. However, numerous smaller supraglacial streams among those large rivers were not delineated. This is exactly the reason why both conservative and non-conservative thresholds are used to constrain the real distribution of supraglacial stream/river networks. This point has now been better explained in the revised manuscript, as requested (p. 9, lines 18-20).

19. ("Page 9, line12: 'mapped rivers' and burned WV DEM.")

**Reply:** Changed as requested (p. 9, line 24).

20. ("Page 9, line 30: How do you define 'large'? A threshold width?")

**Reply:** We defined large supraglacial rivers as the features that can be identified by moderate-resolution (10 – 30 m) satellites (e.g., Sentinel-2 and Landsat-8), while small supraglacial streams as the features that can only be identified by high-resolution (0.5 – 2.0 m) satellites (e.g., WorldView-1/2/3/4). It is subjective to determine a threshold width but if required, we recommend 10 m. This point has been better explained (p. 10, line 12; p. 22, lines 9-12).

21. ("Page 10, line 4 and Table 1: What is 'E'? Please explain.")

**Reply:** E is Nash-Sutcliffe model efficiency (NSE). E is replaced with NSE to for clarity (p. 10, line 17).

22. ("Page 10, lines 8 and 9: I think this is fairly obvious. Suggest rephrase to 'This finding confirms'")

**Reply:** Changed as requested (p. 10, line 21).

23. ("Page 10, section 5.4: What is the 'time to peak' in your catchment? Did you look at this? If not, why not?")

**Reply:** We have reported the "time to peak" in the Rio Behar catchment (~ 6 hours) in Smith et al. (2017). In this study, we reported total supraglacial travel time rather than time to peak. We use this sentence to better distinguish these two related but different parameters.

24. ("Page 11, lines 3 and 4: I think this is significant for broader scale studies where use of a WV DEM is impractical. What about grids of the order of 100 m?")

**Reply:** 100 m or coarser resolution DEM will yield larger offsets in simulating moulin hydrographs compared to 30 m resolution DEM. They will also fail to distinguish between

fine-scale supraglacial streams versus interfluves, the main process-level distinction offered by the RWF routing model. See our reply to comment #1 for more detail (p. 18, lines 30-31; p. 19, lines 1-7).

25. ("Page 11, line 8: Have you tried modifying your SRLF routine to include interfluve flow?")

**Reply:** SRLF assumes the entire ice surface behaves like a supraglacial meltwater channel, and therefore Manning's open-channel flow equation is used to route surface meltwater downslope the bare ice surface to catchment outlet. Therefore, interfluve flow is not included in SRLF. To include interfluve flow, the first step is to partition interfluve and open-channel zones, which is not implemented by SRLF either. If we partition interfluve and open-channel zones and calculate spatially varied velocities for the two zones, we basically create a new spatially-distributed meltwater routing model that is no longer SRLF. See our reply to the first comment of Reviewer 1 for more detail (p. 16, lines 13-31; p. 17, lines 23-25).

26. ("Page 12, line 1: Also earlier in the melt season I expect, i.e. before your study period starts.")

**Reply:** During early melt season, snowpack covers the ice surface and a different method is required to route meltwater within and downslope snowpack. Arnold et al (1998) handled this process but this study only focuses on meltwater routing on bare ice surface.

27. ("Page 12, line 14: Delete repeated 'IDC'")

**Reply:** Deleted as requested (p. 12, line 25).

28. ("Page 13, lines 12-14: How? Would you need proglacial discharge measurements for each catchment?)

**Reply:** We can apply optimal hillslope and open-channel velocities calibrated for the Rio Behar catchment to generate RWF Unit Hydrograph (RWFUH) for other ungauged IDCs. Consequently, moulin discharge hydrograph for each IDC can be estimated by convolving surface melt with RWFUH and the surface routing delays can be calculated from the output moulin hydrographs. These delays can then be integrated into the corresponding RCM grid cells and thus better parameterize surface runoff in RCM simulations. Additional new text has been added to better explain this, as requested (p. 16, lines 24-31).

29. ("Page 15, line 7: Is it possible to characterise surface conditions using your satellite images or would an in-situ investigation be necessary?")

**Reply:** We believe that in-situ investigation is necessary to characterize interfluve conditions. Cooper et al. (2018) analyzed the density and hydrological properties of bare, ablating ice away from open channels, by drilling holes into wet bare ice and measuring the subsurface porosity and water infilling rate, properties that cannot be measured from remote sensing.

Satellite images are certainly useful for providing preliminary observations for ice surface conditions. For example, Smith et al. (2017) partitioned bare ice and snowpack zones using high-resolution satellite imagery and Ryan et al. (2018) investigated ice surface albedo, surface impurities, and cryoconite holes using higher-resolution UAV images. That said, we are unaware of any remote sensing solution to confirm presence/absence of saturated subsurface weathering crust and its hydraulic conductivity, so field measurements remain essential at present.

We have added a new section "6.5 Field site and observation recommendation" to explain how to select field sites and which observations are primarily important for better quantifying surface meltwater routing (p. 18, lines 3-18).

Cooper, M. G., L. C. Smith, A. K. Rennermalm, et al. (2018), Meltwater storage in low-density near-surface bare ice in the Greenland ice sheet ablation zone, Cryosph., 12: 955-970.

Ryan, J. C., A. Hubbard, M. Stibal, et al. (2018), Dark zone of the Greenland Ice Sheet controlled by distributed biologically-active impurities, Nat. Commun., 9(1): 1065.

Smith, L. C., K. Yang, L. H. Pitcher, et al. (2017), Direct measurements of meltwater runoff on the Greenland ice sheet surface, Proc. Natl. Acad. Sci., 114(50): E10622-E10631.

30. ("Figure 1: Explicitly say that the moulin is under the black star.")

**Reply:** Changed as requested (p. 28, Figure 1).

Thank you for considering this manuscript for publication in **The Cryosphere**. If we may provide any additional information about the dataset or analysis, please do not hesitate to contact us via the lead author at kangyang@nju.edu.cn.

Respectfully submitted,

Kang Yang
Associate Professor

[revised manuscript text omitted]